

# Verifying national inventory-based combustion emissions of CO₂ across the UK and mainland Europe using satellite observations of atmospheric CO and CO₂

Tia R. Scarpelli[1,a], Paul I. Palmer[1,2], Mark F. Lunt[1,b], Ingrid Super[3], and Arjan Droste[3,4]

[1]School of GeoSciences, University of Edinburgh, Edinburgh, EH9 3FF, UK
[2]National Centre for Earth Observation, University of Edinburgh, Edinburgh, EH9 3FF, UK
[3]Department of Climate, Air and Sustainability, TNO, P.O. Box 80015, 3508 TA Utrecht, The Netherlands
[4]Department of Water Management, Delft University of Technology, 2628 CN Delft, The Netherlands
[a] Now at: Carbon Mapper, Pasadena, CA, USA
[b] Now at: Environmental Defense Fund, Perth, Australia

*Correspondence to*: Paul I. Palmer (pip@ed.ac.uk)

**Abstract.** Under the Paris Agreement, countries report their anthropogenic greenhouse gas emissions in national inventories, used to track progress towards mitigation goals, but they must be independently verified. Atmospheric observations of $CO_2$,

interpreted using inverse methods, can potentially provide that verification. Conventional $CO_2$ inverse methods infer natural $CO_2$ fluxes by subtracting *a priori* estimates of fuel combustion from the *a posteriori* net $CO_2$ fluxes, assuming that *a priori* knowledge for combustion emissions is better than for natural fluxes. We describe an inverse method that uses measurements of $CO_2$ and carbon monoxide (CO), a trace gas that is co-emitted with $CO_2$ during combustion, to report self-consistent combustion emissions and natural fluxes of $CO_2$. We use an ensemble Kalman filter and the GEOS-Chem atmospheric

transport model to explore how satellite observations of CO and $CO_2$ collected by TROPOMI and OCO-2, respectively, can improve understanding of combustion emissions and natural $CO_2$ fluxes across the UK and mainland Europe, 2018-2021. We assess the value of using satellite observations of $CO_2$, with and without CO, above what is already available from the *in situ* network. Using $CO_2$ satellite observations lead to small corrections to *a priori* emissions that are inconsistent with *in situ* observations, due partly to the insensitivity of the atmospheric $CO_2$ column to $CO_2$ emission changes. When we introduce

satellite CO observations, we find better agreement with our *in situ* inversion and a better model fit to atmospheric $CO_2$ observations. Our regional mean *a posteriori* combustion $CO_2$ emission ranges 4.6—5.0 Gt a$^{-1}$ (1.5—2.4% relative standard deviation), with all inversions reporting an overestimate for Germany's wintertime emissions. Our national *a posteriori* $CO_2$ combustion emissions are highly dependent on the assumed relationship between $CO_2$ and CO uncertainties, as expected. Generally, we find better results when we use grid-scale based *a priori* $CO_2$:CO uncertainty estimates rather than a fixed

relationship between the two species.



## 1 Introduction

More than 40% of the cumulative net $CO_2$ emissions from 1850 to 2019 have occurred since 1990, resulting in a global mean surface temperature rise of 0.45°C (IPCC, 2022). If the 2019 emission rate continues to 2030, we will have exhausted the remaining carbon budget to keep global mean temperatures within 1.5°C and depleted a third of the remaining carbon budget for 2°C (IPCC, 2022). These estimates assume that the land biosphere and ocean will continue to respond to changes in climate as they do today. The most effective lever at our disposal to rapidly reduce atmospheric concentrations of $CO_2$ is a

commensurately large, rapid, and targeted reduction in emissions, as recognized by the Paris Agreement. A clearer understanding of the national importance of individual $CO_2$ emitting sectors is needed to develop effective emission mitigation policies. Similarly, global to regional observing networks are needed to verify the effectiveness of these policies to reduce national emissions from individual sectors. Here, we focus on the potential of satellite observations to verify changes in combustion emissions of $CO_2$ across the UK and mainland Europe.


Under the Paris agreement, countries annually report estimates of their anthropogenic greenhouse gas emissions in national inventories, typically with a lag of more than 12 months, an approach to establish and track progress towards emission mitigation goals. These inventory-based estimates use 'bottom-up' methods that typically rely on national activity data (e.g., power plant fuel consumption) and country-specific emission factors (e.g., $CO_2$ emissions per unit of fuel consumed); the

corresponding emission uncertainties are related to the underlying datasets and methodologies. To set effective national emission mitigation targets and track progress, it is important to estimate $CO_2$ combustion emissions accurately in these inventories, including accurate estimates of their uncertainties.

Observations of atmospheric $CO_2$ provide an independent evaluation of reported bottom-up $CO_2$ flux estimates (e.g., Peylin et

al., 2013). A 'top-down' approach uses these atmospheric measurements to infer the most likely *a posteriori* distribution of $CO_2$ fluxes that would explain the observations, accounting for uncertainties associated with the measurements of the method. An atmospheric transport model is used to relate the gridded *a priori* estimates of $CO_2$ fluxes to 4-D distributions of atmospheric $CO_2$ concentrations. An observation operator is then applied to this 4-D distribution, which describes how a particular instrument samples the atmosphere at a given time and place. The resulting model atmospheric $CO_2$ measurements

are then confronted with the observations, and the *a priori* flux estimates are adjusted to minimize any model-observation differences, resulting in *a posteriori* flux estimates that are consistent with *a priori* and measurement information. Ground-based *in situ* observations from the pan European measurement network have been used extensively to estimate regional net $CO_2$ fluxes (e.g., Scholze et al., 2019, Ramonet et al., 2020; Rödenbeck et al., 2020; Thompson et al., 2020).

Separating the combustion and natural components of the net *a posteriori* $CO_2$ flux estimates is non-trivial, which has resulted in a range of approaches being developed by researchers (e.g., Montanyà et al., 2014; Boschetti et al., 2018; Yang et al., 2023).



The most common approach is to assume we have near-perfect knowledge of anthropogenic emissions, subtract these *a priori* emission estimates from the net *a posteriori* values, and then interpret the residual as fluxes from the natural biosphere to compare with inventory estimates (e.g., White et al., 2019; Deng et al., 2022). The spatial and temporal information on both
emissions and uncertainties are often highly uncertain but also needed to interpret atmospheric measurements (Super et al., 2020, Oda et al, 2023). There is also now a greater focus on estimating changes in anthropogenic emissions, as countries introduce policies to decarbonize their economies.

With this impetus in mind, there is an urgent need to develop and evaluate robust methods that separate the combustion and
natural influences on changes in atmospheric $CO_2$ at city-scale (e.g., Silva et al., 2013; Reuter et al., 2019; Goldberg et al., 2019; Yang et al., 2023) and national-scale (Palmer et al., 2006) using additional observations of trace gases co-emitted during the combustion process. Previous work has focused on using ground-based or aircraft *in situ* measurements of $CO_2$ and co-emitted trace gases, but we need to understand how we best use satellite observations to estimate anthropogenic emissions of $CO_2$, particularly in the context of the billion-euro investment in the Copernicus $CO_2$ Monitoring Mission, CO2M (Sierk et al.,
80  2021).

Observations of atmospheric $CO_2$ collected by satellites have the advantage of global spatial coverage, subject to cloud cover, and have been used to constrain $CO_2$ flux estimates on the spatial scale of 1000s km (e.g., Chevallier et al., 2014; Feng et al., 2017; Chevallier et al., 2019; Palmer et al., 2019). To date, few studies have focused on using these data to constrain $CO_2$ flux
estimates over mainland Europe or the UK because there is less information about surface $CO_2$ on those spatial scales from the current generation of $CO_2$ satellites (OCO-2 and GOSAT) than the *in situ* measurement networks. This is in part because satellite observations of the atmospheric column of $CO_2$ are less sensitive to $CO_2$ surface fluxes compared to *in situ* measurement networks. It is widely anticipated that the significant increase in the volume and spatial coverage of data collected by CO2M will dramatically increase the competitiveness of satellite observations for estimating national-scale emissions
across mainland Europe and the UK.

In this study, we quantify the ability of current satellite observations of $CO_2$ and CO to constrain country-scale combustion and non-combustion $CO_2$ flux estimates across the UK and mainland Europe. We use atmospheric $CO_2$ observations from the NASA OCO-2 instrument and CO observations from the ESA TROPOspheric Monitoring Instrument (TROPOMI) to estimate
monthly $CO_2$ fluxes for 2018-2021. Our work is part of a larger effort to develop rigorous methods to evaluate nationally reported $CO_2$ emissions using *in situ* and satellite observations. In the next section, we describe the methods we use to infer simultaneously combustion and natural fluxes of $CO_2$ using OCO-2 and TROPOMI data. Section 3 describes our results. We conclude the paper in Section 4.



## 2 Data and Methods

Here, we describe the measurements we use to infer $CO_2$ fluxes across the UK and mainland Europe; the GEOS-Chem atmospheric chemistry transport model that describes the relationship between *a priori* inventories, atmospheric chemistry and transport, and the observed atmospheric concentrations of $CO_2$; and the ensemble Kalman Filter that is used to infer $CO_2$ fluxes from *a priori* knowledge and the measurements.

### 2.1 Satellite and *In Situ* Observations

For $CO_2$, we use observations of the atmospheric $CO_2$ column-averaged dry-air mole fraction ($XCO_2$) from the OCO-2 satellite, launched in 2014 (Crisp et al., 2017; Eldering et al., 2017). We use OCO-2 ACOS v10r data for 2018-2021 (OCO-2 Science Team et al., 2020; Taylor et al., 2023). For CO, we use XCO observations from TROPOMI, July 2018 – December 2021, aboard the Sentinel-5P satellite, launched in 2017 (Veefkind et al., 2012; for CO retrieval: Vidot et al., 2012; Landgraf et al., 2016). For both satellite products, we filter observations as recommended in the Product User Guide, including a strict quality assurance flag value of >0.75 for TROPOMI XCO. We remove glint observations and those over the oceans and collate satellite columns and averaging kernels to a 0.25° x 0.3125° spatial grid to match model output (Figure 1). To compare our model output to the satellite observations, we first sampled the model at the overpass time and location of each instrument. We then interpolate our model pressure levels to the satellite pressure levels and apply the scene-dependent retrieval averaging kernel to our 3-D model concentration fields.

We use *in situ* observations for 2018-2021 (Figure 1). We use the DECC surface measurement network in the UK (Stanley et al., 2018; Arnold et al, 2019; O'Doherty et al, 2019a,b) and the ICOS measurement network for Europe (ICOS R., 2022), including drought adjusted observations for 2018 (Ramonet et al., 2020). We retain *in situ* observations collected between 0900 and 1800 local time – to avoid instances when tall tower inlets sit above a shallow boundary layer – and then time-average to 3-hourly intervals to match our GEOS FP model meteorology. All *in situ* sites have $CO_2$ observations, but some sites are missing CO observations. We additionally remove observations when the atmosphere is not well-mixed. We consider the atmosphere to be well-mixed when the standard deviation of $CO_2$ concentrations across the lowest five vertical model levels is smaller than 0.3 ppm.

Figure 1 also shows European sites from the Total Carbon Column Observing Network (TCCON). Five sites are within our domain, including Bremen (Germany; Notholt et al, 2022), Karlsruhe (Germany; Hase et al, 2023), Nicosia (Cyprus; Petri et al, 2022), Orléans (France; Warneke et al, 2022), and Paris (France; Té et al, 2022). We use the TCCON observations as an independent comparison for our inversion results.



## 2.2 Forward Model Description

The forward model **H** describes the relationship between *a priori* flux estimates of $CO_2$ and CO and the atmospheric observations. We use the GEOS-Chem atmospheric chemistry transport model to relate surface fluxes of $CO_2$ and CO to 4-D atmospheric concentrations. We then sample these concentration fields at the time and location of measurements. In the case of satellite observations, we also use the scene-dependent averaging kernel to describe the instrument vertical sensitivity to changes in $CO_2$ and CO. Resulting sampled model atmospheric values can then be compared with observations:

$$\mathbf{y} = \mathbf{H} \cdot \mathbf{x} \quad (1)$$

where **y** denotes the observation vector, and **x** denotes the state vector that includes our *a priori* $CO_2$ and CO flux estimates.

We use the GEOS-Chem version 12.5.2 atmospheric chemistry and transport model which we run at 0.25° x 0.3125° resolution for a nested European domain (-15 to 35° E longitude and 34 to 66° N latitude) with 47 vertical levels. GEOS-Chem is driven by GEOS FP meteorological re-analyses fields from the NASA Global Modelling and Assimilation Office (GMAO) Global Circulation Model.

Our *a priori* flux estimates (**x**) include all sources contributing to observed atmospheric $CO_2$ and CO. Equation 2 shows the sources for $CO_2$ including combustion emissions ($CO_2^{Combust}$), non-combustion fluxes (both biogenic and non-combustion anthropogenic sources; $CO_2^{Bio}$), and background $CO_2$ that is transported to and from our domain ($CO_2^{Trans}$). Atmospheric CO sources include combustion emissions ($CO^{Combust}$), transport ($CO^{Trans}$), and production of CO through oxidation ($CO^{Chem}$), as shown in equation 3.

$$CO_2 = CO_2^{Trans} + CO_2^{Combust} + CO_2^{Bio} \quad (2)$$
$$CO = CO^{Trans} + CO^{Combust} + CO^{Chem} \quad (3)$$

For our 2018-2021 *a priori* fluxes, we use a combination of regional and global inventories (Figure 2). Combustion emissions for both species ($CO_2^{Combust}$ and $CO^{Combust}$) are from the TNO GHGco v5.0 emission inventory at 0.1° x 0.05° resolution (Super et al., 2020; Kuenen et al., 2022) with national totals based on emissions reported in national inventories and extrapolated from 2019 to more recent years. We apply scaling factors provided by TNO to reflect monthly, hourly, and daily patterns in emissions by sector with the same scaling factors used for each year. Our combustion source also includes biomass burning emissions from the GFAS v1.2 inventory (Kaiser et al., 2021). Non-combustion fluxes ($CO_2^{Bio}$) include ocean fluxes from the NEMO-PISCES model (Lefèvre et al., 2020), lateral carbon fluxes related to crop removal (Deng et al., 2022), and hourly terrestrial biosphere fluxes at 1/120° x 1/60° resolution produced by the VPRM model following methods described by Gerbig



(2021) driven by ERA5 meteorology. We include non-combustion (e.g., fugitives) anthropogenic emissions from the TNO inventory in our non-combustion fluxes.


For our nested domain, we use boundary conditions for $CO_2$ ($CO_2^{Trans}$) from the CAMS inversion-optimized global greenhouse gas analysis with assimilation of *in situ* observations (Chevallier, 2020). Our boundary conditions for CO ($CO^{Trans}$) are from the CAMS global reanalysis (Inness et al., 2019). We use the CAMS fields at their provided temporal resolution (3-hourly) and re-grid to the GEOS-Chem horizontal spatial resolution of 2° x 2.5°. Because the vertical resolution of GEOS-Chem does

not align with CAMS, we translate the CAMS native vertical resolution to our 47 model layers using linear interpolation of logarithmic pressure values. We fill in the species concentrations at the lowest or highest pressure level in CAMS for the top or surface of the atmosphere, respectively, when the GEOS-Chem pressure levels go beyond the bounds of CAMS.

We treat the relationship between surface fluxes and concentrations (equation 1) as linear (e.g., a doubling of emissions leads

to a doubling of the atmospheric signal). To linearize the CO simulation, we use offline chemistry terms to represent the chemical production of CO ($CO^{Chem}$). CO is primarily produced by oxidation of methane and non-methane volatile organic compounds by the hydroxyl radical (OH), so we generate the production terms using offline 3-D loss fields of OH generated from a previous GEOS-Chem full-chemistry simulation (Fisher et al., 2017).

**2.3 Inverse Model Description**

For our inversion, we use the Ensemble Kalman Filter (EnKF) approach as discussed in detail by others (e.g., Peters et al., 2005; Hunt et al., 2007; Feng et al., 2009; Liu et al., 2016). We specifically follow the methods derived by Hunt et al. (2007) and summarized by Liu et al. (2016) for the Local Ensemble Transform Kalman Filter (LETKF).

We solve the inversion in ensemble space rather than for the state vector elements. For each state vector element, we have an

ensemble of potential scale factors that follow our prescribed error statistics. For each assimilation time-period (over which we ingest observations), we solve for the mean *a posteriori* state vector ($\bar{\mathbf{x}}^a$) that represents the mean of our $N$ ensemble members (where we use $N = 100$):

$$\bar{\mathbf{x}}^a = \bar{\mathbf{x}}^b + \mathbf{K}(\mathbf{y}_{obs} - \bar{\mathbf{y}}^b), \quad (4)$$


where $\bar{\mathbf{x}}^a$ and $\bar{\mathbf{x}}^b$ are the means across ensemble members for our *a posteriori* and *a priori* state vectors, respectively. We use error statistics, as described in Section 2.4, to generate the *a priori* state vector ensemble members. $\mathbf{y}_{obs}$ is the observation vector and each element of $\bar{\mathbf{y}}^b$ is the mean of model-predicted concentrations across $N$ ensemble members. For the $n$th ensemble member ($\mathbf{x}_n^b$), the model-predicted concentrations are $\mathbf{y}_n^b = H(\mathbf{x}_n^b)$.




**K** describes our Kalman gain matrix that regulates the degree to which any disagreement between model and observation will adjust the state vector. We determine **K** using the matrix $\mathbf{X}^b$, which describes the difference between the ensemble members and their mean, and the matrix $\mathbf{Y}^b$, which describes the difference between the model-predicted concentrations and their mean:

$\quad \mathbf{K} = \mathbf{X}^b \widetilde{\mathbf{P}}^a (\mathbf{Y}^b)^{\mathrm{T}} \mathbf{R}^{-1}$, (5)

where the $n$th column of $\mathbf{X}^b$ is $\mathbf{x}_n{}^b - \bar{\mathbf{x}}^b$ and the $n$th column of $\mathbf{Y}^b$ is $\mathbf{y}_n{}^b - \bar{\mathbf{y}}^b$ (each column representing an ensemble member). **R** is the observation error covariance matrix, which includes the errors from our forward model and observations. For $CO_2$, we use an *a priori* model error of 1.5 ppm for the satellite inversion (Feng et al., 2017) and 3 ppm for the *in situ*

inversion (within the range of Monteil et al., 2020 and White et al., 2019). For CO, we use an *a priori* model error of 15 and 20 ppb for the satellite and *in situ* inversions, respectively (Northern Hemisphere CO column and surface mole fraction model-observation differences from Bukosa et al., 2023). For the observations, we use the errors as provided for the satellite or *in situ* network, averaged to the model resolution. We generate the off-diagonal covariance for **R** based on the spatial and temporal proximity of observations following an exponential decay with spatial and temporal length scales of 100 km and 4 hours,

respectively.

The $\widetilde{\mathbf{P}}^a$ matrix is a representation of the *a posteriori* error covariance in ensemble space:

$\widetilde{\mathbf{P}}^a = [(N-1)\mathbf{I} + (\mathbf{Y}^b)^{\mathrm{T}} \mathbf{R}^{-1} \mathbf{Y}^b]^{-1}$, (6)


where **I** is an identity matrix and $N$ is our number of ensemble members. $\widetilde{\mathbf{P}}^a$ is used to determine the *a posteriori* ensemble members ($\mathbf{X}^a$) where the $n$th column of $\mathbf{X}^a$ is $\mathbf{x}_n{}^a - \bar{\mathbf{x}}^a$ and the error covariance matrix ($\mathbf{P}^a$):

$\mathbf{X}^a = \mathbf{X}^b [(N-1)\widetilde{\mathbf{P}}^a]^{1/2}$ (7)

$\quad \mathbf{P}^a = \mathbf{X}^a (\mathbf{X}^a)^T (N-1)^{-1}$. (8)

We use an assimilation window of two weeks and a lag window of one month, accounting for the impact of historical emissions on our assimilation period. This means that the state vector for each time-step includes scale factors for the assimilation window and lag window. We perform our inversion sequentially, using the *a posteriori* scale factors for a given assimilation window

to update the *a priori* scale factors for the next lag window over the same date range. To avoid unrealistically small prior uncertainties, we apply a 10% error inflation when we update the *a priori* state vector.



The benefit of the LETKF is that we can localize the inversion so that each state vector element is only influenced by a subset of observations. For our inversions using *in situ* observations, we localize by distance so that each state vector element that
represents a grid-scale variable is only influenced by observations within a 1000 km range.

**2.4 Description of Inverse Model Experiments**

We test different approaches to investigate the usefulness of satellite observations for evaluating $CO_2$ combustion emissions. The approaches vary in the observations that are used and the representation of error covariances for our *a priori* estimates. For each type of inversion, we compare our satellite inversion results to comparable inversions using *in situ* observations.

In the inversions, instead of solving for $CO_2$ or CO fluxes, we solve for scale factors that scale up or scale down the source terms from equations 2-3. We first assume that our *a priori* scale factors are all equal to one. We solve for *a posteriori* scale factors that, when applied to our source terms, will result in modelled atmospheric $CO_2$ or CO concentrations in better agreement with observations.

For our first approach ($CO_2$-only), we perform a $CO_2$-only inversion that assimilates $CO_2$ observations. Our state vector includes scale factors for the sources of equation 2:

$$\mathbf{x}_{co2} = \left(\mathbf{x}_{co2}{}^{Trans}, \mathbf{x}_{co2}{}^{Bio}, \mathbf{x}_{co2}{}^{Combust}\right) \quad (9)$$


where $\mathbf{x}_{co2}{}^{Bio}$ and $\mathbf{x}_{co2}{}^{Combust}$ are a vector of scalers with each element applying to a non-combustion or combustion grid cell at $0.5° \times 0.625°$ resolution (Appendix A). For the transport scale factors, each element of $\mathbf{x}_{co2}{}^{Trans}$ applies to $CO_2$ transported from the North, South, East, or West boundary.

In our second approach (Joint $CO_2$:CO), we perform a joint $CO_2$:CO inversion that assimilates both $CO_2$ and CO observations. For the joint inversion, we assume there is 100% correlation for the $CO_2$ and CO combustion emission errors. This means any adjustment made by our inversion to the $CO_2$ combustion scale factors will also apply to the CO scale factors and vice versa. We can then use a common combustion scaling term for both species in our state vector ($\mathbf{x}^{Combust}$). Our state vector also includes scale factors for transport of each species (i.e., allowing adjustment of our assumed background concentration), and
for CO we include a vector with two scale factors for the chemistry terms ($\mathbf{x}_{co}{}^{Chem}$):

$$\mathbf{x}_{co2} = \left(\mathbf{x}_{co2}{}^{Trans}, \mathbf{x}_{co2}{}^{Bio}, \mathbf{x}^{Combust}\right) \quad (10)$$
$$\mathbf{x}_{co} = \left(\mathbf{x}_{co}{}^{Trans}, \mathbf{x}_{co}{}^{Chem}, \mathbf{x}^{Combust}\right). \quad (11)$$



For our first two approaches, we assume an *a priori* uncertainty of 20% (relative standard deviation) for the combustion scale factors ($\mathbf{x}^{Combust}$). We use an *a priori* uncertainty of 50% for the non-combustion scale factors ($\mathbf{x}_{co2}{}^{Bio}$), and 5% for the atmospheric transport and chemistry scale factors. For our non-combustion and combustion scale factors, we generate error covariances for nearby grid cells that exponentially decays with increasing distance. Our method for generating the error covariance matrix based on these uncertainties is described in detail in Appendix A.


We acknowledge that the assumption of 100% error correlation for $CO_2$ and CO combustion emissions is likely to be a gross overestimate. For example, we may underestimate CO emissions due to an underestimate of incomplete combustion activities, and this will not translate to the same underestimate in $CO_2$.

For our third approach (TNO $CO_2$:CO), we test this assumption by solving for the $CO_2$ and CO combustion scaling terms separately:

$$\mathbf{x}_{co2} = \left(\mathbf{x}_{co2}{}^{Trans}, \mathbf{x}_{co2}{}^{Bio}, \mathbf{x}_{co2}{}^{Combust}\right) \quad (12)$$
$$\mathbf{x}_{co} = \left(\mathbf{x}_{co}{}^{Trans}, \mathbf{x}_{co}{}^{Chem}, \mathbf{x}_{co}{}^{Combust}\right) \quad (13)$$


We call this our TNO approach because we use estimates of the uncertainties in the TNO emission inventory to create our error covariance matrix (Super et al., 2024). We increase the provided uncertainties by a factor of 3 to make them more comparable with our other simulations. This results in a mean grid-scale $CO_2$ combustion uncertainty of 18%, though there is greater variability in grid cell uncertainties compared to our other approaches. We expect higher correlation between $CO_2$ and

CO gridded emissions in regions where the same spatial product is used to distribute emissions for both species (e.g., road network maps) and that spatial product has high uncertainties.

## 3 Results and Discussions

First, we describe the comparison between our *a priori* and *a posteriori* model simulations against observations. We then report our *a posteriori* $CO_2$ fluxes for Europe and its constituent countries and the UK.

**3.1 Inversion Performance**

Our *a priori* $CO_2$ emissions are already consistent with data from the five relevant TCCON sites (locations shown in Figure 1; Pearson correlation coefficient R= 0.87), and *in situ* (R=0.76) and satellite (R=0.84) observations. The model has a small, positive relative mean bias compared to TCCON (0.7%) and a very small bias compared to *in situ* and satellite observations (0.2%). Table A1 reports a statistical summary of the model-observation comparisons. The satellite inversions show



improvement for the model-satellite fit (R=0.92-0.95), as expected, and the model-*in situ* fit (R=0.80-0.82). Similarly, the *in situ* inversions improve model-*in situ* fit (R=0.83-0.84) and to a lesser extent the model-satellite fit (R=0.85-0.87).

In general, including CO and TNO uncertainty estimates improves the model-observation fit and reduces the mean bias. For example, the satellite joint $CO_2:CO$ (R=0.93) and TNO (R=0.92) inversions show the greatest improvement in fit with TCCON.

The one exception is that the mean bias compared to TCCON is slightly larger with CO (0.3-0.5%) compared to $CO_2$-only (0.2-0.4%). The TCCON $CO_2$ bias is seasonal with the *a priori* model showing no bias in July-August and a positive bias of 1-4 ppm for the rest of the year. The *in situ* inversions reduce the mean bias for March-June by 1 ppm, and this improvement lines up with a reduction in the biosphere sink for these inversions (discussed later).

We also assess inversion performance by the degree of uncertainty reduction for the *a posteriori* $CO_2$ combustion emission estimates. Table 1 shows *a posteriori* uncertainties for our domain-scale $CO_2$ combustion emissions. The reductions in relative uncertainty achieved at the domain scale for all inversions are small (6-12%) with the $CO_2$-only and TNO satellite inversions showing no reduction. The TNO inversions show smaller reductions in uncertainty (0-6%) compared to the joint inversions (8-12%), but they also start with a lower *a priori* uncertainty at 1.6% (relative standard deviation; RSD) compared to 2.4% for

non-TNO *a priori* uncertainties.

At the national scale, we see the greatest uncertainty reduction in $CO_2$ combustion emissions for the top 10 emitting countries when satellite CO observations or *in situ* $CO_2$ measurements are included and the non-TNO uncertainties are used (Table A2 and A3). The average uncertainty reductions for the joint satellite and $CO_2$-only *in situ* inversions are 11% and 9%,

respectively. This is not surprising given the greater number of observations provided by these two platforms and increased sensitivity to surface fluxes compared to OCO-2. Including *in situ* CO observations in the inversion does not improve the national-scale uncertainty reduction. Because we use lower *a priori* uncertainties in the TNO inversion (national-scale 2-10% RSD) compared to the other inversions (6-14% RSD), fewer countries have reduced uncertainties for the TNO inversion, though *a posteriori* uncertainties are reduced in the Netherlands (2%) for both in-situ and satellite compared to *a priori*

uncertainties (3%).

### 3.2 Emission Estimates for the UK and Mainland Europe

Table 1 shows our mean domain-scale (includes the UK and mainland Europe) combustion emissions for 2018-2021. The inversions show a small decrease or no change from the *a priori* emissions (4.9 Gt a$^{-1}$), except for the joint satellite and *in situ* inversions that show a larger decrease (4.6 Gt a$^{-1}$) and an increase (5.0 Gt a$^{-1}$) from the *a priori*, respectively. Figure 3 shows

that the joint satellite inversion decreases combustion emissions year-round for all years with the greatest decreases in winter.



The TNO satellite/*in situ* and $CO_2$-only *in situ* inversions also show decreases in the winter and early spring (Figures 3 and 4), providing more confidence in this scaling down of emissions.

In contrast, the joint *in situ* inversion shows an increase for all months and all years (Figure 4). This pattern is not reflected in our other inversion approaches and is likely, in part, due to the model underestimating the fine-scale variability in CO compared to what is measured at some *in situ* sites combined with the use of a common scale factor for both CO and $CO_2$, leading to an over-correction upward of combustion emissions. For example, we find that removing a single site close to an urban region in northern Italy (Ispra ICOS site) reverses the sign of scaling in the region from an increase to a decrease. The disagreement between satellite and *in situ* $CO_2$:CO inversions is less pronounced for the TNO inversions because the separation of $CO_2$ and CO in our state vector prevents the CO underestimates from heavily influencing the $CO_2$ combustion emissions.

Figure 3 shows a slight (1%) decrease in mean *a priori* combustion emissions from 2018 to 2021, and all satellite and *in situ* inversion results show a similar trend (Figure 3 and 4). The mean *a priori* non-combustion (biogenic) $CO_2$ sink shows a slight increase (1%) for 2018-2021, and the inversion results show a similar (satellite; Figure 3) or greater increase (*in situ*; Figure 4*)* in the $CO_2$ sink. Figure 4 shows the monthly mean biogenic $CO_2$ sink is weakened for the *in situ* inversions, mostly in summer, whereas Figure 3 shows almost no change in the sink for the satellite inversions (also listed in Table A4), indicating that the $CO_2$ *in situ* observations, due to the coverage and sensitivity they provide, are needed for constraining biogenic flux estimates.

The differences between *a posteriori* and *a priori* annual emissions for all inversions except the joint satellite inversion are not statistically significant and remain within the 1-σ uncertainties of the *a priori* estimate. The inter-annual trends are also smaller in magnitude than the *a posteriori* uncertainties, making it difficult to assess if $CO_2$ combustions in Europe have decreased from 2018 to 2021. For the joint satellite and *in situ* inversions, we assumed that CO was a strong tracer for $CO_2$ combustion emissions on this regional scale by using a common scale factor, but we find that this assumption leads to more extreme, likely unrealistic, divergence from the *a priori*, in disagreement with the other inversion results. This reflects the difficulties of using CO as a tracer for $CO_2$ combustion emissions at regional scales, and the importance of error characterization.

### 3.2 National-scale Emission Estimates

Figure 5 shows national $CO_2$ combustion emissions for the top 10 emitting countries in our European domain (also listed in Tables A2 and A3). Germany is the highest emitter with an *a priori* emission of 821 Tg a⁻¹. Most inversions show a decrease in Germany's emissions (717-806 Tg a⁻¹) except for the *in situ* joint inversion which shows an increase (830 Tg a⁻¹) and the $CO_2$-only inversion which shows little change from the *a priori* estimate (819 Tg a⁻¹). The other top emitting countries, including Poland, the UK, France, Italy, Spain, Belgium, the Czech Republic, the Netherlands, and Romania, show emission decreases for the satellite joint (3-17%) and TNO (0-4%) inversions. The *in situ* $CO_2$-only and TNO inversions generally show



only small changes (<1%) in national emissions except for a 4% national emission decrease in the Netherlands and Belgium

for the $CO_2$-only inversion.

The joint inversions show the largest changes in national emissions but in opposite directions. In contrast, the TNO inversions show smaller changes from the *a priori* (in part, due to the lower *a priori* uncertainties) and better agreement, including agreement in Germany where there is greater divergence from the *a priori* estimate (2% decrease for both TNO inversions).


Despite the national-scale disagreements for some inversions, we find regional corrections to combustion emissions are consistent for all inversions. Figure 6 shows that the populated North Rhine-Westphalia region in western Germany shows a decrease in $CO_2$ combustion emissions for all inversions. The TNO and $CO_2$-only inversions show mixed corrections in Poland with TNO inversions showing the best agreement. Most inversions, including both TNO inversions, show an increase in

emissions near Milan and Vienna, but over other major cities like Paris, Madrid, and London there is less agreement in the sign and magnitude of the emissions changes.

The differences in the joint inversions are due to contrasting corrections to CO emissions that carry over into the $CO_2$ emissions. Figures A1 and A2 show that the *in situ* joint inversion shows decreases for high-emitting regions in Europe for winter and

spring, but this is mostly offset by large emission increases in summer and fall. In contrast, the satellite joint inversion shows decreases for all seasons. For the TNO inversion, there is less disagreement between the seasonal emissions corrections for $CO_2$, but there are disagreements in CO corrections. Figure A3 shows the CO corrections for the TNO inversion generally occur at the national-scale and we know there is low error correlation between the two species at the national-scale (Super et al., 2024), so it is not surprising that these corrections do not carry over to $CO_2$.


Figure 7 shows national non-combustion (biogenic) emissions for the countries in Figure 5. All countries show a net sink with France having the largest net sink. The *in situ* inversions tend to decrease (lessen) the $CO_2$ sink for all countries and reduce uncertainties. Figure 8 shows the spatial pattern in the flux changes is consistent for all *in situ* inversions. In contrast, the national $CO_2$ biogenic fluxes show little change from the *a priori* for the satellite inversions, highlighting the importance of *in*

*situ* $CO_2$ observations for constraining biogenic flux estimates. For all inversions, the $CO_2$ sink in northern Germany is strengthened (more negative fluxes) and weakened in southern Germany and Switzerland, though there are conflicting corrections in surrounding regions such as France and northern Italy. These disagreements may be due to the differing observing capacities with satellites having seasonal limitations due to snow and clouds. We find low *a posteriori* error



correlations between national-scale combustion and biogenic fluxes (mostly R<0.1, except for Germany R=-0.2), indicating
that the disagreement in *in situ* and satellite *a posteriori* biogenic fluxes will not carry over into combustion emission estimates.

## 4 Conclusions

We find that using $CO_2$ satellite observations from OCO-2 alone cannot reproduce *a posteriori* European CO2 fluxes inferred from the European *in situ* $CO_2$ measurement network. The satellite observations (CO2-only) do not show significant combustion emissions changes from our a priori estimates, whereas when we use *in situ* $CO_2$ or $CO_2$ and CO satellite
observations, we see greater divergence from the a priori emissions. We find that the *in situ* network is still essential for constraining biogenic fluxes, though we also find low correlation between combustion and biogenic fluxes indicating that our inability to constrain the biogenic flux estimate using satellites does not prevent the estimation of combustion emissions at the national scale using satellite observations.

All our inversions indicate that $CO_2$ combustion emissions for regions of Germany are overestimated in winter, and most inversions show this overestimate extends to other countries in Europe. We also find that the *in situ* inversions show a smaller summertime European $CO_2$ sink which is not shown for the satellite inversions. We find that the existing observational networks are not able to significantly reduce the errors for our European or national emission estimates to the extent necessary for distinguishing inter-annual emission trends that represent only a few percent of total emissions.


When using CO as a tracer for $CO_2$ combustion emissions in our inversion system, we find that our interpretation of inversion results is highly dependent on the assumptions of *a priori* error correlation between CO and $CO_2$. The use of a CO:$CO_2$ inversion system can potentially improve our ability to track CO2 combustion emissions provided we have well-characterized error correlations between the two species which may require broad measurement based studies to determine the error
correlations specific to a source and region. This suggests that the increase in observational capacity for $CO_2$ and co-emitted trace gases promised by the Copernicus CO2 Monitoring (CO2M) satellite mission has the potential to improve our ability to constrain national combustion emission estimates provided that the error correlations for $CO_2$ combustion emissions and the co-emitted species are strong and well characterized using empirical data.

In general, the improvements in model-observation fit are small and we do not see significant reduction in uncertainties compared to our *a priori* estimate. This is expected because we have extensive knowledge about sector emissions that underpin these regional inventories. The use of CO observations and TNO error estimates leads to better agreement between satellite and *in situ* inversions and the best model-observation fit, though including CO does not reduce the model bias compared to TCCON and likely reflects the need for *in situ* $CO_2$ observations for reducing biases related to biogenic fluxes. Despite the
sensitivity of our *a posteriori* emission estimates to the choice of *a priori* $CO_2$ and CO uncertainties, the joint and TNO satellite



inversions perform similarly when compared to TCCON. This highlights the need for not only further satellite observing capacity but also improved ground-based networks for evaluating satellites and the usefulness of including co-emitted species observations.

**Code and data availability**

The community-led GEOS-Chem model of atmospheric chemistry and transport model is maintained centrally by Harvard University (https://geoschem.github.io/, last access: 15 February 2024), and is available on request. The ensemble Kalman filter code is publicly available at https://github.com/mflunt/enkf-code (last access 15 February 2024). The L2 column carbon dioxide data from OCO-2 and OCO-3 are available from the Goddard Earth Sciences Data and Information Services Centre (https://disc.gsfc.nasa.gov/datasets; last access 15 February 2024). The Sentinel-5P TROPOMI column methane and carbon monoxide data are available from the Copernicus Data Space Ecosystem (https://dataspace.copernicus.eu/; last access 15 February 2024).  The TCCON data were obtained from the TCCON Data Archive hosted by Caltech DATA at https://doi.org/10.14291/TCCON.GGG2020 (TCCON Team, 2022), with individual station DOIs cited in the main text. ICOS data are available from the data portal (https://data.icos-cp.eu/; last access 15 February 2024), with the data release DOI cited in the main text.

**Author contribution**

TRS and PIP designed the research; TRS prepared the calculations, with help from MFL on the inverse method; IS and AD provided the error correlation data and expert advice on its usage, with inputs from TRS and PIP; TRS and PIP wrote the paper, with inputs from MLF, IS, and AD.

**Competing interests**

The authors declare that they have no conflict of interest.

**Acknowledgements**

This research was supported by the European Commission, Horizon 2020 Framework Programme VERIFY (grant agreement # 776810 for Paul Palmer and Mark Lunt) and CoCO2 (grant agreement # 958927 for Tia Scarpelli, Paul Palmer, Ingrid Super, and Arjan Droste). Paul Palmer also received support from UK National Centre for Earth Observation funded by the Natural Environment Research Council (grant # NE/R016518/1). We thank the OCO-2 and TROPOMI satellite retrieval teams, the European TCCON leads, and the UK (DECC) and mainland European (ICOS) *in situ* data providers. We also thank the GEOS-Chem community, particularly the team at Harvard University who help to maintain the GEOS-Chem model, and the NASA Global Modeling and Assimilation Office (GMAO) who provided the MERRA2 data product.



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



**Figures**

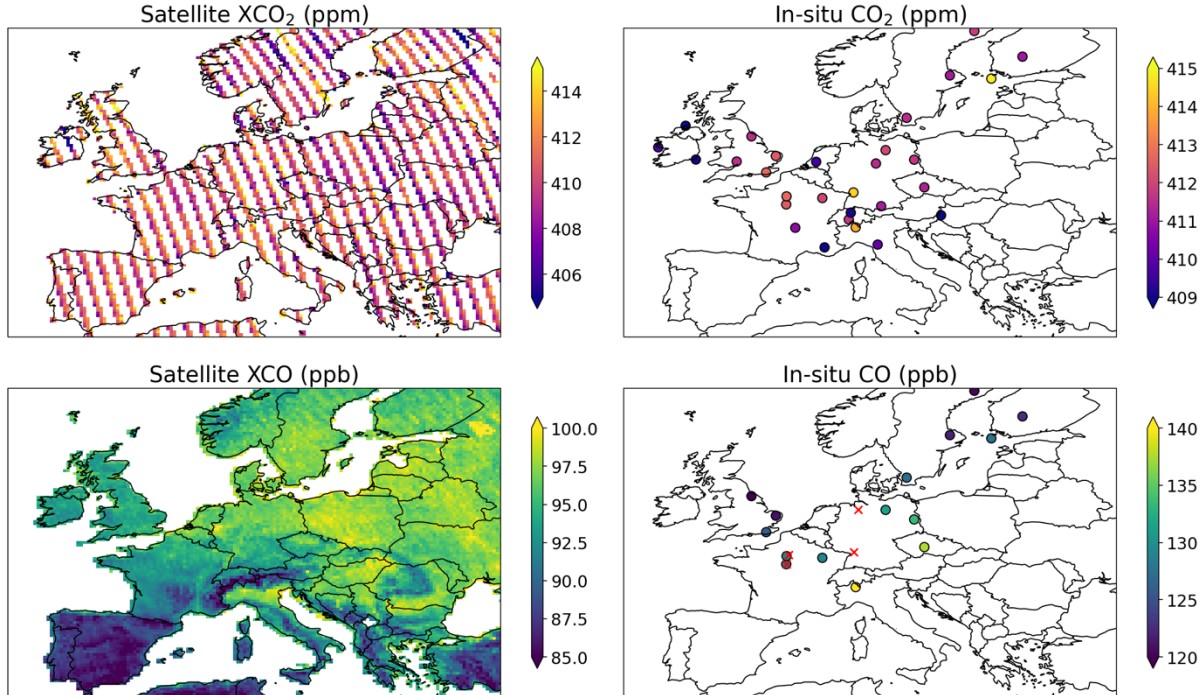

**Figure 1**. Annual mean CO$_2$ and CO observed by satellite and *in situ* networks across Europe for 2018-2021. Satellite observations of XCO$_2$ and XCO are from OCO-2 and TROPOMI, respectively, and *in situ* observations are from the DECC and ICOS networks. The red X points in the *in situ* CO plot show the locations of the five TCCON sites we used to evaluate

our inversions. The observations are filtered as stated in the text and satellite observations are shown at 0.25° x 0.3125° resolution. TROPOMI observations only include observations after July 2018.





**Figure 2.** Annual mean emissions for 2018-2021 in the *a priori* inventories. Combustion emissions ($CO_2^{combust}$, $CO^{combust}$) are from the TNO inventory while biogenic fluxes ($CO_2^{bio}$) are from the VPRM model (negative values indicate a $CO_2$ sink).



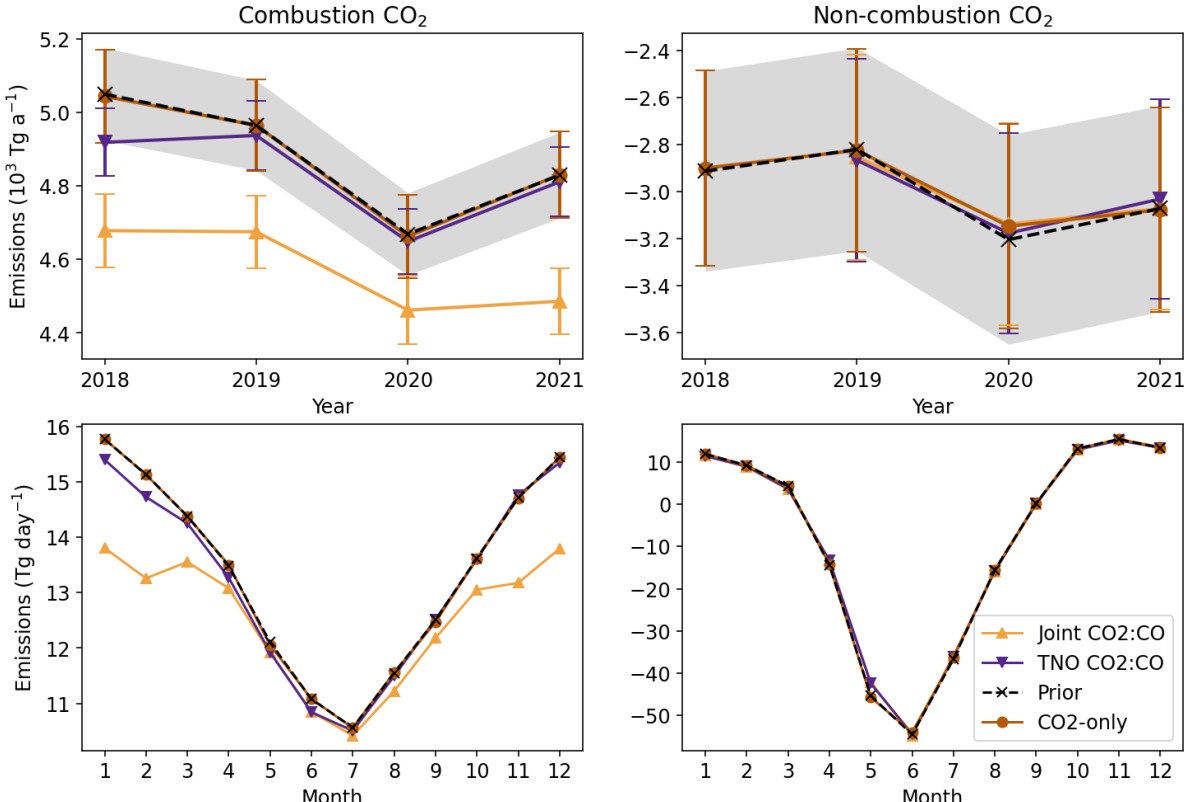

**Figure 3.** Annual and monthly mean European CO₂ combustion and non-combustion emissions inferred from satellite inversions for 2018-2021. The non-combustion emissions include biogenic and non-combustion anthropogenic emission sources. The top row shows annual mean CO₂ flux estimates by inversion type, with errors bars showing the 1-σ errors except for the *a priori* errors which are shown as a shaded region. The bottom row shows monthly mean fluxes for 2018-2021. The TNO and joint inversions only include July 2018-December 2021 for combustion and 2019-2021 for non-combustion.



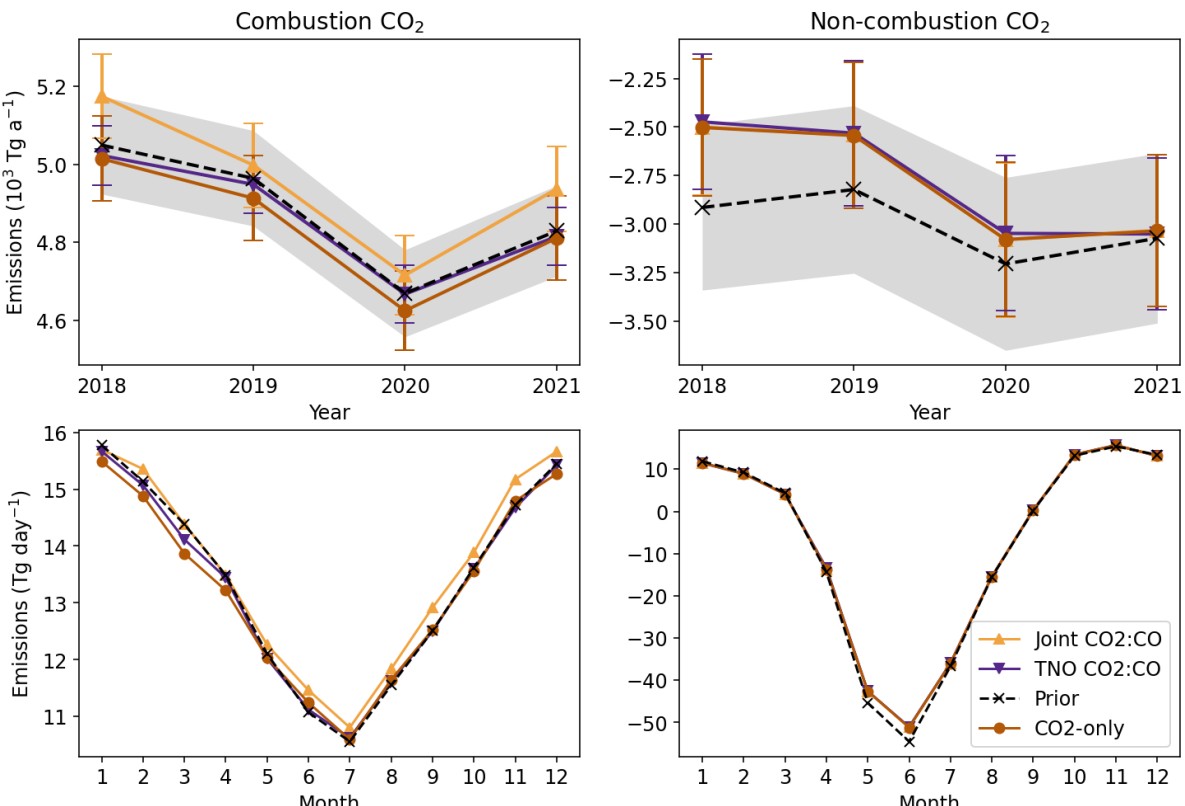

**Figure 4.** The same as Figure 4 for *in situ* inversions.



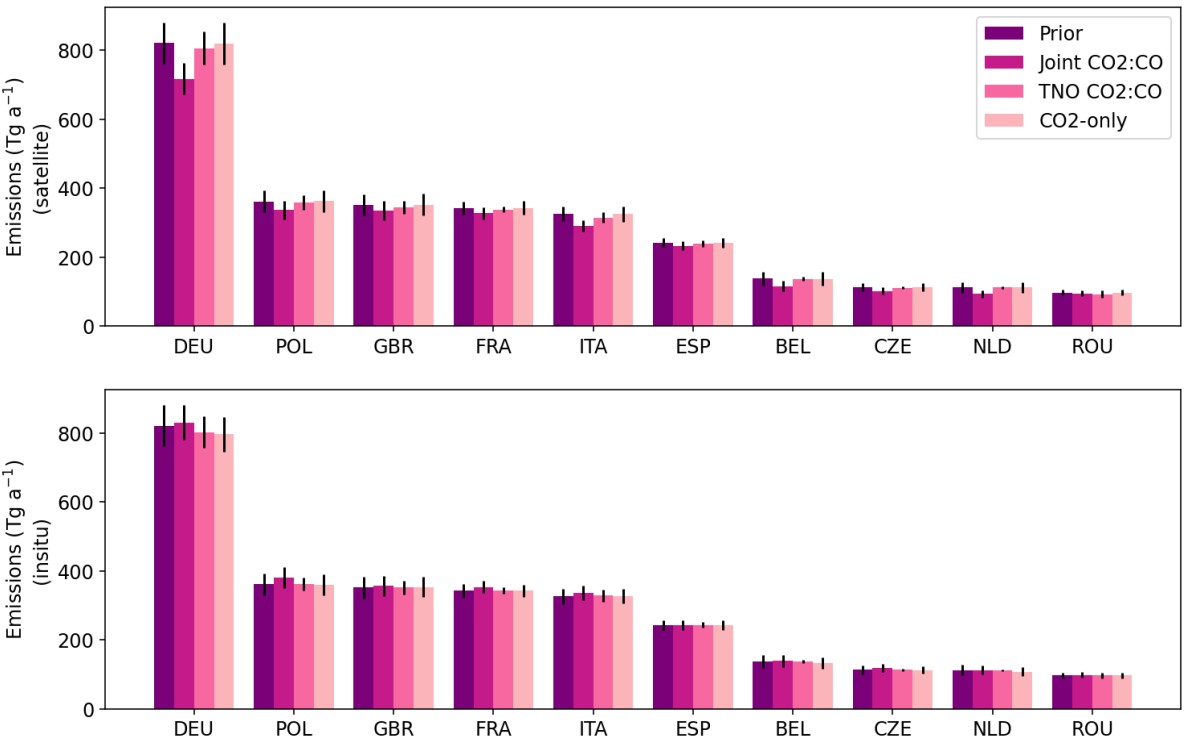

**Figure 5.** Annual mean *a priori* and *a posteriori* $CO_2$ combustion emissions by country for satellite (top) and *in situ* (bottom) inversions. We show the top 10 emitting countries in our European domain with emissions averaged over 2018-2021. The

TNO and joint satellite inversion averages do not include dates prior to July 2018.



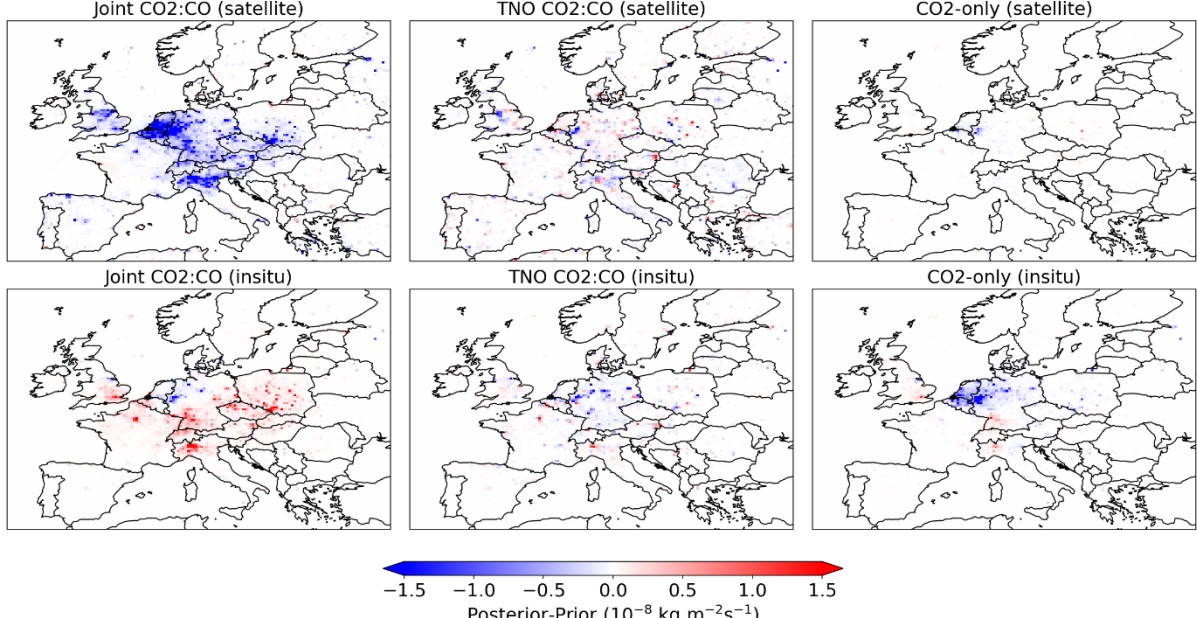

**Figure 6.** Annual mean $CO_2$ combustion emissions difference (*a posteriori* minus *a priori*) for satellite (top row) and *in situ* (bottom row) inversions, 2018-2021, shown at the native model resolution of 0.25° x 0.3125°. The TNO and joint satellite inversion averages do not include dates prior to July 2018.





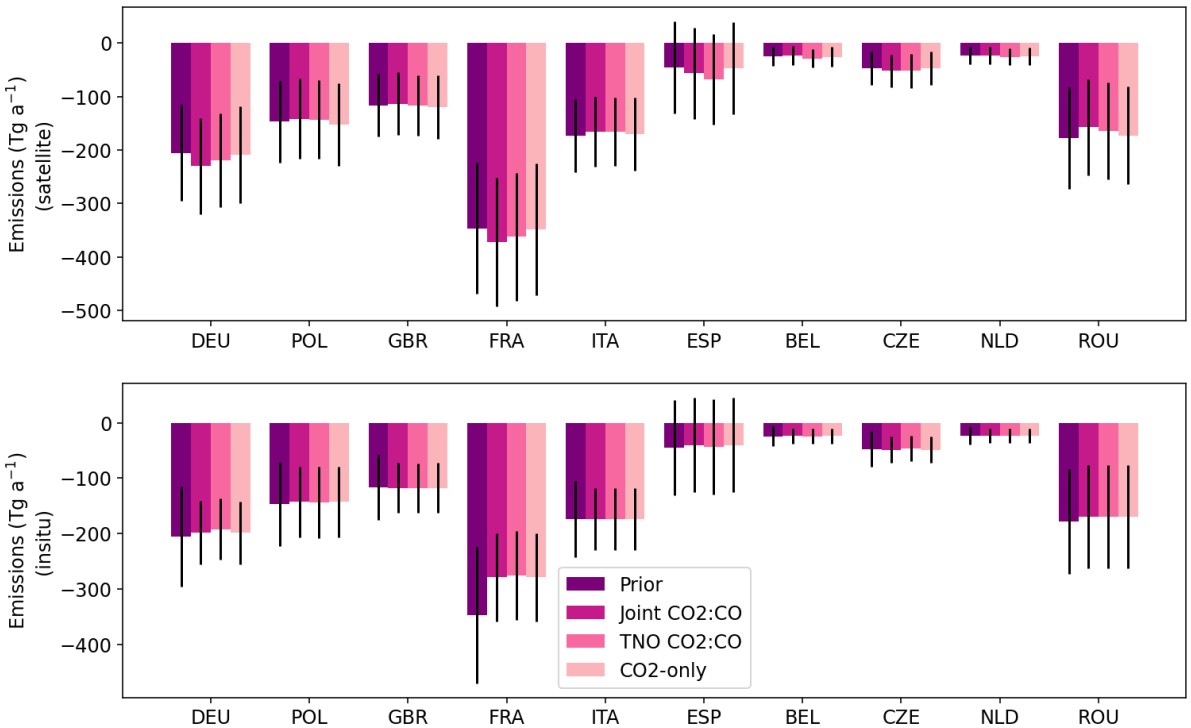

**Figure 7.** As Figure 5 but for non-combustion CO$_2$ fluxes estimates. The TNO and joint satellite inversion averages do not include 2018. The non-combustion emissions include biogenic and non-combustion anthropogenic emission sources.




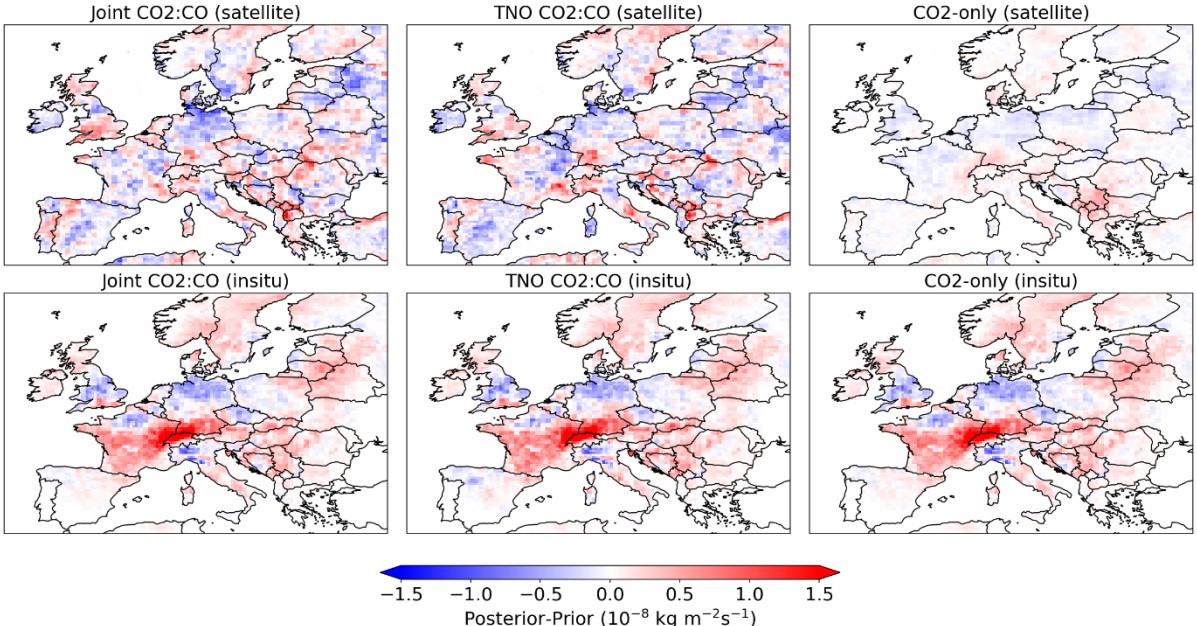

**Figure 8.** As Figure 6 but for non-combustion $CO_2$ flux estimates. The TNO and joint satellite inversion averages do not include 2018. The non-combustion emissions include biogenic and non-combustion anthropogenic emission sources.






**Tables**

**Table 1**. Average domain $CO_2$ combustion emissions for 2018-2021

| | Mean (Gt a$^{-1}$) | | RSD [a] |
|---|---|---|---|
| | Emission | Change [a] | (%) |
| *A priori*[b] | 4.9 | | 2.4 |
| TNO *a priori*[b] | 4.9 | | 1.6 |
| Satellite | | | |
|     $CO_2$-only | 4.9 | - | 2.4 |
|     Joint $CO_2$:CO[c] | 4.6 | ▼ | 2.1 |
|     TNO $CO_2$:CO[c] | 4.8 | - | 1.6 |
| In-situ | | | |
|     $CO_2$-only | 4.8 | ▼ | 2.2 |
|     Joint $CO_2$:CO | 5.0 | ▲ | 2.2 |
|     TNO $CO_2$:CO | 4.9 | - | 1.5 |

680   [a]The arrows indicate the change of the mean from the *a priori*. Blue-downward pointing arrows show a decrease, red-upward show an increase, and grey dashes show no change. RSD stands for relative standard deviation.
[b]The *a priori* uncertainty labelled as '*A priori*' is for the $CO_2$-only and joint inversions, so we also include the *a priori* uncertainty for the TNO inversion.
[c]The Joint and TNO satellite inversions only include July 2018 - December 2021. The *a priori* combustion emission for this
685   period is 4.8 Gt a$^{-1}$ so we show no change for the TNO *a posteriori* emissions.



**Appendix A**

**1 Description of an *a priori* ensemble generation**

For the *a priori* ensemble perturbations that represent our state vector ($\mathbf{x}_n^b$ for the *n*th ensemble member), we generate an ensemble of scale factors based on the desired error statistics, described in Section 2.3. For the combustion and non-combustion scale factors, we solve for scale factors on a 0.5° x 0.625° resolution grid (double our nested model resolution). Each ensemble member is then a grid of perturbations that we will apply to our emissions grid. To generate the ensemble members, we first generate an error covariance matrix (**P**):

$$\mathbf{P} = \mathbf{P}' \cdot \left( e^{-\frac{\mathbf{D}}{100}} \cdot \mathbf{P}' \right),$$

where $\mathbf{P}'$ is a diagonal matrix with the variance for each state vector element along its diagonal. Covariance between grid cells is based on the spatial proximity between each grid cell and its neighbor with distances between grid cells represented by the matrix **D**. The influence of neighboring grid cells decreases with distance following an exponential decay with a length scale of 100 km.

We then perform a Cholesky decomposition on **P**. We generate each *a priori* ensemble member by applying a random perturbation vector with mean zero and standard deviation equal to one ($\boldsymbol{\eta}$) to the decomposed matrix (**L**) and adding one (which is the assumed *a priori* mean of all ensemble members):

$$\mathbf{x}_n^b = 1 + \mathbf{L} \cdot \boldsymbol{\eta}.$$

The TNO emissions inventory is constructed by allocating national emissions to a grid using a spatial map of activity data (e.g., a population map), so the uncertainty in the gridded emission estimate is a combination of the uncertainty in the national emissions and the uncertainty in the spatial product used to distribute emissions. We use uncertainty estimates for the national emissions, uncertainties for the spatial products, and estimates of the correlation of uncertainties for the two species to generate an ensemble of gridded emissions by sector, following a Monte Carlo approach. This method is described in Super et al. (2023). We use the emissions ensemble to generate an error covariance matrix (**P**) and follow the steps outlined in the main text.






**Table A1.** Fit of *a priori* and *a posteriori* modelled $CO_2$ compared to observations for 2018-2021[a]

| | Correlation coefficient | | | Relative mean bias (%) | | |
|---|---|---|---|---|---|---|
| | *In situ* | TCCON[b] | Satellite | *In situ* | TCCON | Satellite |
| *A priori* | 0.76 | 0.87 | 0.84 | 0.2 | 0.7 | 0.2 |
| Satellite inversions | | | | | | |
| $\quad$ $CO_2$-only | 0.81 | 0.90 | 0.92 | -0.2 | 0.4 | -0.1 |
| $\quad$ Joint $CO_2$:CO[c] | 0.80 | 0.93 | 0.95 | -0.2 | 0.5 | <0.05 |
| $\quad$ TNO $CO_2$:CO[c] | 0.82 | 0.92 | 0.95 | -0.1 | 0.5 | <0.05 |
| *In situ* inversions | | | | | | |
| $\quad$ $CO_2$-only | 0.83 | 0.85 | 0.85 | -0.4 | 0.2 | -0.2 |
| $\quad$ Joint $CO_2$:CO | 0.84 | 0.85 | 0.86 | -0.3 | 0.4 | -0.1 |
| $\quad$ TNO $CO_2$:CO | 0.84 | 0.86 | 0.87 | -0.3 | 0.3 | -0.1 |




[a]We use the Pearson's correlation coefficient and relative mean bias (the means of the *a posteriori* and *a priori* difference divided by the *a priori*) as measures of fit.

[b]Five sites are within our domain (Figure 2) include Bremen (Germany), Karlsruhe (Germany), Nicosia (Cyprus), Orléans (France), and Paris (France).




**Table A2**. Annual mean national $CO_2$ combustion emissions (Emis; Tg a$^{-1}$) and relative standard deviations (RSD; %) for 2018-2021 satellite inversions

| Country | Country Abbr. | Prior | | CO₂-only | | Joint[a] | | TNO[a,b] | | |
|---|---|---|---|---|---|---|---|---|---|---|
| | | Emis | RSD | Emis | RSD | Emis | RSD | Emis | RSD | PRSD |
| Germany | DEU | 821 | 7 | 819 | 7 | 717 | 6 | 806 | 6 | 6 |
| Poland | POL | 361 | 9 | 362 | 9 | 336 | 8 | 358 | 5 | 5 |
| United Kingdom | GBR | 351 | 9 | 352 | 9 | 335 | 8 | 345 | 5 | 6 |
| France | FRA | 342 | 6 | 343 | 6 | 327 | 5 | 338 | 2 | 2 |
| Italy | ITA | 326 | 7 | 325 | 7 | 291 | 6 | 314 | 5 | 5 |
| Spain | ESP | 242 | 6 | 242 | 6 | 233 | 5 | 239 | 4 | 4 |
| Belgium | BEL | 137 | 14 | 137 | 14 | 116 | 13 | 136 | 4 | 4 |
| Czech Republic | CZE | 113 | 11 | 113 | 11 | 102 | 10 | 111 | 3 | 3 |
| Netherlands | NLD | 112 | 13 | 112 | 13 | 93 | 12 | 112 | 2 | 3 |
| Romania | ROU | 97 | 8 | 97 | 8 | 95 | 8 | 93 | 10 | 10 |

[a]The satellite inversions that include CO only show means for July 2018 - December 2021.

[b]The *a priori* uncertainties for TNO differ from the CO₂-only and joint inversions, so we list the TNO *a priori* uncertainties (PRSD) as well. The higher *a posteriori* error for Romania is due to the error inflation factor used in the sequential inversion.



**Table A3.** Annual mean national $CO_2$ combustion emissions (Emis; Tg a$^{-1}$) and relative standard deviations (RSD; %) for 2018-2021 *in situ* inversions[a]

| Country | Country Abbr. | CO₂-only | | Joint | | TNO | |
|---|---|---|---|---|---|---|---|
| | | Emis | RSD | Emis | RSD | Emis | RSD |
| Germany | DEU | 796 | 6 | 830 | 6 | 802 | 6 |
| Poland | POL | 360 | 8 | 380 | 8 | 361 | 5 |
| United Kingdom | GBR | 353 | 8 | 356 | 8 | 352 | 6 |
| France | FRA | 342 | 5 | 353 | 5 | 342 | 2 |
| Italy | ITA | 327 | 6 | 336 | 6 | 328 | 5 |
| Spain | ESP | 243 | 6 | 243 | 6 | 243 | 3 |
| Belgium | BEL | 132 | 13 | 139 | 12 | 137 | 4 |
| Czech Republic | CZE | 112 | 10 | 119 | 10 | 113 | 3 |
| Netherlands | NLD | 107 | 12 | 112 | 12 | 112 | 2 |
| Romania | ROU | 97 | 8 | 98 | 8 | 97 | 9 |

[a]Only the *a posteriori* emissions are shown. The *a priori* emissions and uncertainties are listed in Table A2.






**Table A4**. Domain mean $CO_2$ non-combustion emissions for 2018-2021[a]

|  | Mean ($Gt\ a^{-1}$) | RSD (%) |
| --- | --- | --- |
| *A priori* | -3.0 | 14 |
| Satellite |  |  |
| $CO_2$-only | -3.0 | 14 |
| Joint $CO_2$:$CO^b$ | -3.0 | 14 |
| TNO $CO_2$:$CO^b$ | -3.0 | 14 |
| In-situ |  |  |
| $CO_2$-only | -2.8 | 14 |
| Joint $CO_2$:CO | -2.8 | 14 |
| TNO $CO_2$:CO | -2.8 | 14 |




[a] The non-combustion emissions include biogenic and non-combustion anthropogenic emission sources.

[b]Joint and TNO inversion satellite results only include 2019-2021. The *a priori* non-combustion flux is the same for this period ($-3.0\ Gt\ a^{-1}$).




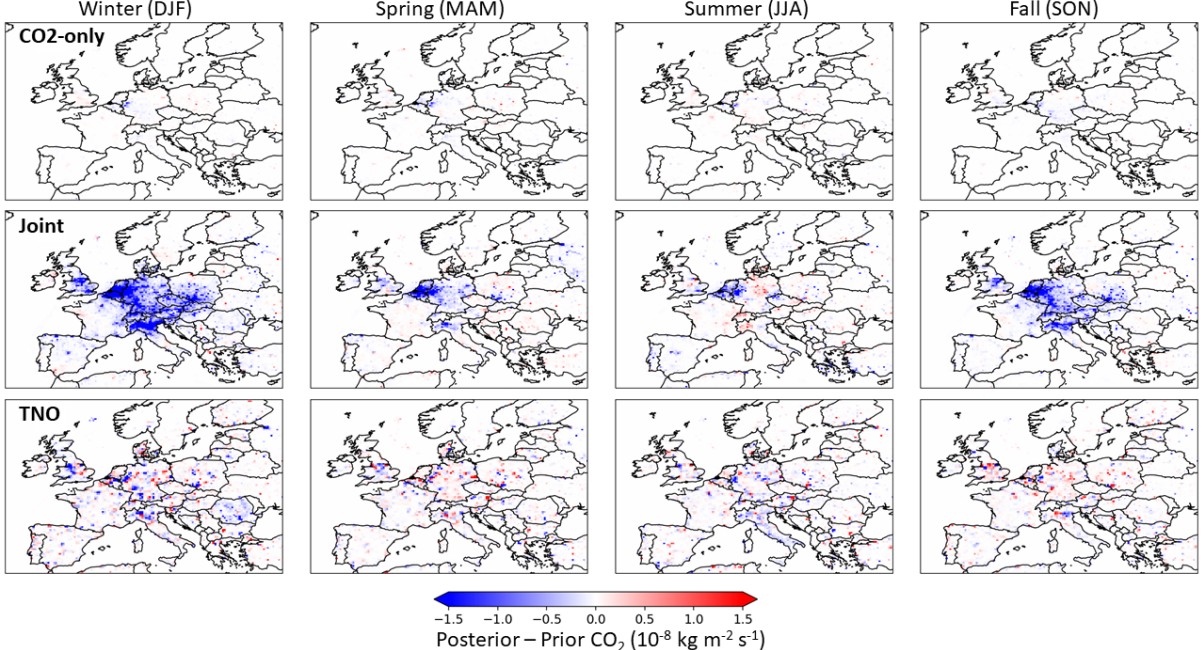

**Figure A1**. Seasonal mean *a posteriori* and *a priori* $CO_2$ combustion emissions difference for satellite inversions for 2018-2021. The inversions including CO satellite observations do not include emissions differences prior to July 2018.






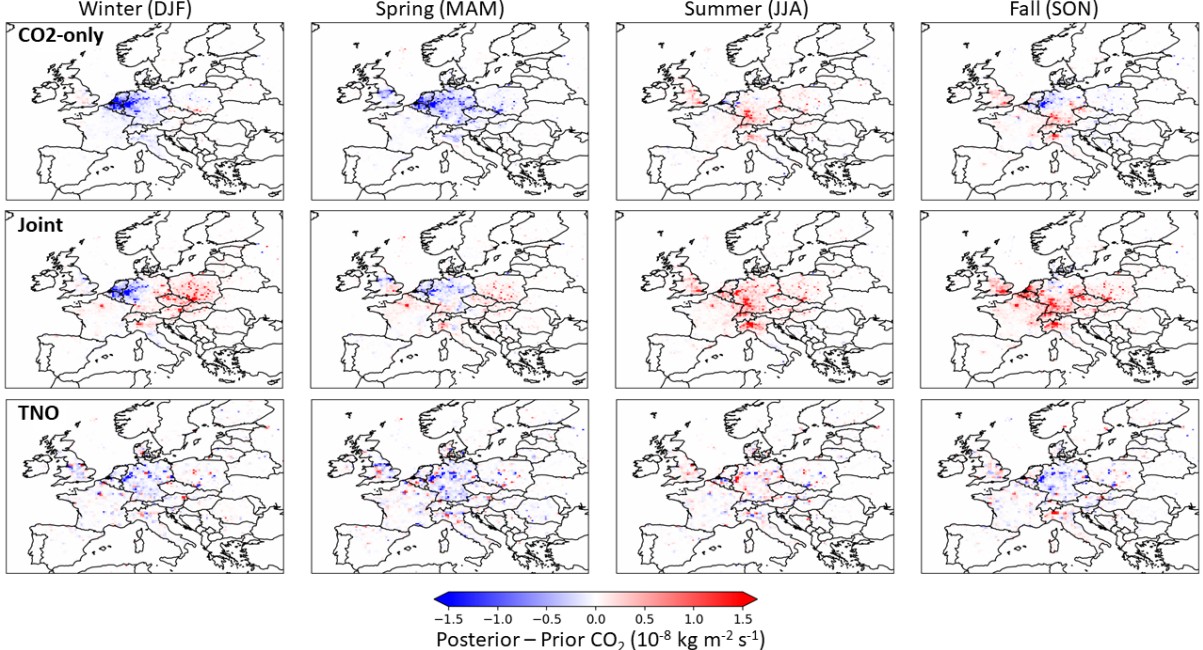

**Figure A2**. Same as Figure A1 for *in situ* inversions.



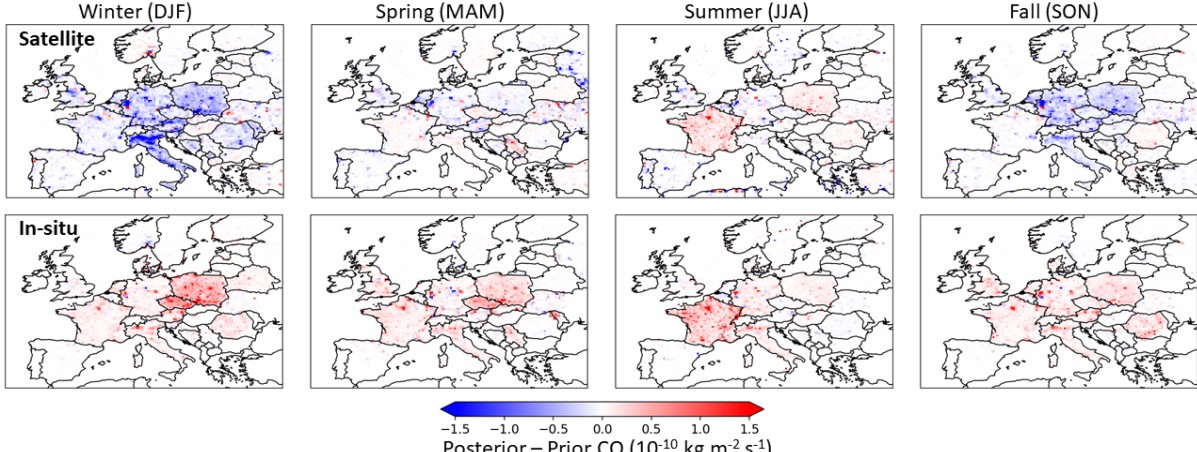


**Figure A3**. Same as Figure A1 for CO in satellite and *in situ* inversions.