# Peer review of "Verifying national inventory-based combustion emissions of CO2 across the UK and mainland Europe using satellite observations of atmospheric CO and CO2"

_EGUsphere, 2024_

## Referee Comment (RC2)

Review on **Verifying national inventory-based combustion emissions of CO2 across the UK and mainland Europe using satellite observations of atmospheric CO and CO2** by Scarpelli et al., 2024

The manuscript presents $CO_2$ emissions at European scale investigating the advantage of using additional CO observations in $CO_2$ satellite and in-situ inversions. The manuscript is well written and relevant for this journal, even though there are missing information and interpretation of the results which would help other readers to fully appreciate this study.

Minor comments are listed below.

General revisions:

Ln. 89. Could you add further information on CO2M such as it spatial and temporal resolution?

Ln. 112 Figure 1. Based on your Figure1, it seems that you are using TCCON data only for CO, but in your manuscript, you mentioned that all TCCON sites have $CO_2$ measurements. So, you could maybe add the red X points also for the in-situ $CO_2$ figure. The northern TCCON site in Germany seems to have no CO and $CO_2$ measurements, can you explain why there are no values for $CO_2$? Additionally, your caption mentioned 5 TCCON sites used to evaluate your inversions, but only 4 X points are shown in your Figure.

Ln. 119. You mentioned using drought adjusted observations for 2018 without further information. Can you develop why you used it and what it consists exactly?

Ln. 124. You only considered observations for a well-mixed atmosphere using a threshold value for the standard deviation of 0.3ppm. How did you estimate this threshold?

Ln. 127. You mentioned using Nicosia TCCON site located in Cyprus, however based on your Figure 1, this TCCON site is outside your domain. Did you use it consequently suggesting your domain is larger than shown in Figure 1, or did you not use it?

Ln. 130. Section 2.2. It is not clear if you did two separate CO and $CO_2$ inversions or if you did a joint inversion. It is not explicitly mentioned and would help the reader if it was mentioned at the beginning of this section.

This information can only be found later in your manuscript. Additionally, you do not provide information on how you treat averaging kernel information and the difference in vertical sensitivity between CO and $CO_2$ measurements to your joint inversion and TNO inversion. Can you give further information? Particularly, do you think that the difference in vertical sensitivity and variation in AVK between CO and $CO_2$ could impact your $CO_2$:CO inversions?

Ln. 156. TNO GHG should be more introduce. What TNO stands for? Why do you need to extrapolate data from 2019?

Ln. 163. Can you give examples of fugitives/non-combustion anthropogenic emissions? These examples could appear in the manuscript ln.163.

Ln.169. You re-grid CAMS fields to the GEOS-Chem horizontal spatial resolution of 2x2.5, this is in contradiction with your information line.141 where you mentioned using GEOS-Chem at 0.25x0.3125 resolution.

Ln. 255. Are the 2 scale factors for the chemistry terms linked to the oxidation of $CH_4$ and NMVOC? If that is the case, that should be clear here. If that is not the case, then further details should be mentioned here for these 2 scale factors.

Ln. 250 – 260. In your second approach of $CO_2$:CO joint inversion, how do you account for $CO_2$ production from the CO reaction with the radical OH?

Ln. 295. Any explanations why the bias is larger with CO than with the $CO_2$-only inversion?

It is nowhere mentioned which emissions are optimized in your inversions. Are the emissions from biomass burning optimized as well as biogenic and anthropogenic emissions?

Ln. 296. Based on which Figure or results do you observed seasonal biases?

Figure 3. Satellite joint inversion shows lower combustion emissions in winter and fall than other months and compared to the other inversions. When looking at Figure A.3, we do observe lower posterior emissions than prior emissions for the same months particularly with satellite observations. Any assumptions why? Are these lower posterior emissions linked to combustion or non-combustion?

Figure 3 and 4. The spatial distribution of observations between satellite and in-situ measurements is not the same over Europe with in-situ observations mainly in northern Europe, however your European a posteriori combustion emission seems to match between both set of observations. How do you explain it? And how would this spatial difference impact your inversions?

Ln. 324. This sentence would need further information and details. I do not see for the joint in situ inversion an increase for all months and all years in Figure 4. Which increase are you talking about? Like all inversions, we can observe a decrease from January to July and then an increase for the rest of the months. Annual and seasonal variabilities seem to agree between all inversions. We do observe an over-estimation for the join inversion at the annual scale compared to the other inversions but not an increase.

Ln. 363. Are we talking about decrease in combustion emissions or an under-estimation/reduction in the estimation of the inversions compared to the prior

emissions? It is confusing to talk about decrease or increase through the results section as it feels there was increase/decrease at national scale through the study period. Results do not suggest that country have reduce/increase their emissions but more that prior estimations are either under-estimated or over-estimated compared to optimized satellite/in-situ emissions.

Ln. 377. Any assumptions why France is showing the largest net sink for both satellite and in-situ inversions compared to other countries?

Technical revisions:

Ln. 81. Among the references cited here, I would suggest adding the MIP studies assimilating $CO_2$ satellites and in-situ measurements in an ensemble of several atmospheric inversions (Crowell et al., 2019 ( https://doi.org/10.5194/acp-19-9797-2019) and Peiro et al., 2022 (https://doi.org/10.5194/acp-22-1097-2022 )). Both studies have used OCO-2 measurements and have looked at Europe emissions among several other regions.

Ln. 92. Carbon monoxide is not introduce in the introduction.

Ln. 403. CO2 should be $CO_2$

Ln. 402 through 405. This sentence is a bit too long which makes it difficult to understand. I would suggest re-writing it.

---

## Author Comment (AC1)

We thank the three reviewers for providing comments on our manuscript that have helped to clarify our key messages. Below we respond to all individual reviewer comments (denoted in italics).

**Reviewer 1**

**General Comments:**

This manuscript presents one of few studies investigating the value of joint constituent inverse analysis of satellite observations in verifying and improving the accuracy of CO2 emissions estimates. This paper is therefore relevant for publication in this journal. However, there are sections of the manuscript that need further clarification and elaboration (both methodological and interpretation of results). For this reason, the reviewer recommends minor revisions for this manuscript. Please see specific comments for details of these concerns.

**Specific Comments:**

1. The analysis performance on CO emissions (chem and trans) is unclear. Is the fit to CO observations also improved? Are the adjustments in posterior CO reasonable? While Figure A3 is informative, it would strengthen the paper if some discussion is added on this aspect.

Posterior improvements of CO against TROPOMI and in situ data were slightly larger than for CO2, reflecting the larger associated assumed errors. We found the posterior CO agreed better for the "Joint" and "TNO" inversions, with the TNO inversion marginally better. We have now added these points to the manuscript.

It is also not quite apparent whether the state vector includes grid-scale scaling factors for each sector (CO2: combust, trans, bio; CO: combust, trans, chem). Also, at what horizontal grid resolution are these state vectors (0.25deg x 0.3125deg or 2 deg x 2.5 deg or 0.5 deg x 0.625 deg)? Please clarify. How many elements are in the state vector xb and data y\_obs?

For the joint CO:CO2 inversion, the state vector includes grid-based (0.5x0.625 resolution) scalars for combustion (CO and CO2) and the natural biosphere (CO2).

The transport scalars correspond to the four lateral boundary conditions. The two chemistry scalars correspond to domain-wide factors applied to the methane and NMVOC fields. The emission scalers by sector and by gas described at the  $0.5 \times 0.625$  degree grid-scale resolution. This information is now clarified in the revised manuscript.

3. Is it not clear why the off-diagonal elements of R is generated? As stated in Line 202, R includes the errors from our forward model and observations. Can R be explicitly calculated from the ensemble?

The only off-diagonal elements of R that we include describe correlations between measurements, determined using an exponential decay based on characteristic spatial and temporal length scales. These are described in section 2.3.

4. While the authors acknowledge that assuming 100% error correlation for CO2:CO is a gross estimate, it is not realistic, and results are therefore not useful and perhaps can be misleading.

We agree. We use this calculation as an illustrative upper limit. We have now emphasized in the text that we use this calculation for illustrative purposes only.

5. The paper states: "The satellite observations (CO2-only) do not show significant combustion emissions changes from our a priori estimates, whereas when we use in situ CO or CO2 and CO satellite observations, we see greater divergence from the a priori emissions." Can this divergence be due to model issues related to representing vertical mixing processes as well?

On reflection, this is more likely to do with a combination of in situ observations being more sensitive to emissions and there being insufficient satellite observations to update the state vector. We have now made this clearer in the revised manuscript.

6. Also, the paper states: "improvements in model-observation fit are small and we do not see significant reduction in uncertainties compared to our a priori estimate." Is this because of lack of information content in the data (accuracy and precision)? Or as the succeeding statements alluded to, that the errors specified in the a priori is already low in the first place. What about biomass burning, which has relatively larger uncertainties for both CO2 and CO?

Based on our analysis, we believe that this is due primarily to the very small uncertainties assigned to the prior emission estimates, unrealistic or otherwise.

Most combustion over Europe is not associated with wildfire emissions on an annual basis, otherwise, we agree, the prior would have larger uncertainties.

- 7. What is the basis of the following statements:
- Line 122. "We consider the atmosphere to be well-mixed when the standard deviation of CO2 concentrations across the lowest five vertical model levels is smaller than 0.3 ppm." Why 0.3?

Our choice of 0.3 is a subjective choice, based on three times the typical in situ measurement precision. We have now explained this is the revised manuscript.

• Line 208. "We generate the off-diagonal covariance for 3 based on the spatial and temporal proximity of observations following an exponential decay with spatial and temporal length scales of 100 km and 4 hours, respectively." Why 100 km and 4 hours?

These values are based on our prior work that showed spatial and temporal model error correlations to be of the order of 100 km and 4 hours.

• Line 221. "We use an assimilation window of two weeks and a lag window of one month?" Why 2 weeks and 1 month?

We used two weeks to account for temporal variations in emissions. Our previous tests showed that lag periods longer than a month didn't have a major impact because signals have substantially decayed by that time and, if anything, may introduce spurious correlations. We now add this clarification in the revised manuscript.

• Line 224. "We perform our inversion sequentially, using the a posteriori scale factors for a given assimilation window to update the a priori scale factors for the next lag window over the same date range." Is the prior error covariance also updated?

No, the prior error covariance is not updated.

• Line 226. "We apply a 10% error inflation when we update the a priori state vector." Why 10%? Is it possible to show Chi-Square statistics?

Our choice of 10% was a subjective choice to avoid an overly certain covariance that would not otherwise allow updates from the ingestion of further data. Given the modest improvements to the state vector, showing chi-square statistics will not add much to the discussion.

 Line 229. "We localize by distance so that each state vector element that represents a grid-scale variable is only influenced by observations within a 1000 km range." Why 1000 km? And does it mean that >1000 km has zero influence? Is there a smooth function that is applied?

Using observations further away than 1000 km may lead to spurious correlations that impact results, so we used a hard cut-off. Our previous tests showed that observations in the immediate vicinity of state vector elements have the most influence. This is explained by typical emission signals decaying exponentially with distance from the point of release.

• Line 253. "Our state vector also includes scale factors for transport of each species (i.e., allowing adjustment of our assumed background concentration), and for CO we include a vector with two scale factors for the chemistry terms (x\_CO^chem)." Why 2 for chem?

The two chemistry scalars correspond to domain-wide factors applied to the methane and NMVOC fields. Methane and NMVOCs have very spatial distribution reflecting differences in their sources and their atmospheric lifetime.

Line 260. "For our first two approaches, we assume an a priori uncertainty of 20% (relative standard deviation) for the combustion scale factors xcombust). We use an a priori uncertainty of 50% for the non-combustion scale factors xco2bio), and 5% for the atmospheric transport and chemistry scale factors." Please justify the choice of these numbers.

These values are informed estimates based on our previous studies. We have now made that clear in the revised manuscript.

 Line 276. "We call this our TNO approach because we use estimates of the uncertainties in the TNO emission inventory to create our error covariance matrix (Super et al., 2024). We increase the provided uncertainties by a factor of 3 to make them more comparable with our other simulations. This results in a mean grid-scale CO2 combustion uncertainty of 18%, though there is greater variability in grid cell uncertainties compared to our other approaches. We expect higher correlation between CO2 and CO gridded emissions in regions where the same spatial product is used to distribute emissions for both species (e.g., road network maps) and that spatial product has high uncertainties." What are these spatial products? Please elaborate.

These spatial products are described in Kuenen et al. 2022 (ESSD - CAMS-REG-v4: a state-of-the-art high-resolution European emission inventory for air quality modelling (copernicus.org). We have re-arranged the statement so it is now clearer.

• Line 699. "The influence of neighboring grid cells decreases with distance following an exponential decay with a length scale of 100 km." Is this assuming isotropy?" If so, is this justifiable?

Yes, and yes. Ultimately, we must make a number of simplifying (and commonly used) assumptions to ensure our calculations are computationally tractable.

**Reviewer 2**

The manuscript presents CO2 emissions at European scale investigating the advantage of using additional CO observations in CO2 satellite and in-situ inversions. The manuscript is well written and relevant for this journal, even though there are missing information and interpretation of the results which would help other readers to fully appreciate this study.

Minor comments are listed below.

General revisions: Ln. 89. Could you add further information on CO2M such as it spatial and temporal resolution?

Good idea. Additional text has now been added to the revised manuscript.

Ln. 112 Figure 1. Based on your Figure1, it seems that you are using TCCON data only for CO, but in your manuscript, you mentioned that all TCCON sites have CO2 measurements. So, you could maybe add the red X points also for the in-situ CO2 figure. The northern TCCON site in Germany seems to have no CO and CO2 measurements, can you explain why there are no values for CO2? Additionally, your caption mentioned 5 TCCON sites used to evaluate your inversions, but only 4 X points are shown in your Figure.

We have addressed this point by revising the Figure 1 caption.

Ln. 119. You mentioned using drought adjusted observations for 2018 without further information. Can you develop why you used it and what it consists exactly?

This was a misunderstanding on our part. The data have not been "adjusted" in any way beyond normal calibration procedures. We have clarified this in the revised manuscript.

Ln. 124. You only considered observations for a well-mixed atmosphere using a threshold value for the standard deviation of 0.3ppm. How did you estimate this threshold?

Our choice of 0.3 is a subjective choice, based on three times the typical in situ measurement precision. We have now explained this is the revised manuscript.

Ln. 127. You mentioned using Nicosia TCCON site located in Cyprus, however based on your Figure 1, this TCCON site is outside your domain. Did you use it consequently suggesting your domain is larger than shown in Figure 1, or did you not use it?

We do use this site, but this Figure is a zoomed-in version of our study domain. We have addressed this comment by revising the Figure 1 caption.

Ln. 130. Section 2.2. It is not clear if you did two separate CO and CO2 inversions or if you did a joint inversion. It is not explicitly mentioned and would help the reader if it was mentioned at the beginning of this section. This information can only be found later in your manuscript. Additionally, you do not provide information on how you treat averaging kernel information and the difference in vertical sensitivity between CO and CO2 measurements to your joint inversion and TNO inversion. Can you give further information? Particularly, do you think that the difference in vertical sensitivity and variation in AVK between CO and CO2 could impact your CO2:CO inversions?

There are differences in the averaging kernels of CO and CO2, but both have sensitivity to changes in the two gases in the lower troposphere. These differences are considered within our inversion framework that applies scene-dependent averaging kernels to the model. We have now clarified this point about the inversions earlier in the manuscript.

*Ln.* 156. TNO GHG should be more introduce. What TNO stands for? Why do you need to extrapolate data from 2019?

TNO is in English: Netherlands Organisation for Applied Scientific Research. We have included this in the manuscript.

Emissions from 2019 is the latest year available for our European emission inventory that includes air pollutants and greenhouse gases. We included a reference to Kuenen et al 2022 that describes the inventory.

*Ln.* 163. Can you give examples of fugitives/non-combustion anthropogenic emissions? These examples could appear in the manuscript In.163.

In case of CO2 and CO these are small. An example from agriculture includes CO2 that escapes from glasshouses enriched with CO2. We have included this example in the revised manuscript.

Ln.169. You re-grid CAMS fields to the GEOS-Chem horizontal spatial resolution of 2x2.5, this is in contradiction with your information line.141 where you mentioned using GEOSChem at 0.25x0.3125 resolution.

We use the coarser scale model output as boundary conditions from the nested, finer scale model calculations. We clarifu this point in the revised manuscript.

Ln. 255. Are the 2 scale factors for the chemistry terms linked to the oxidation of CH4 and NMVOC? If that is the case, that should be clear here. If that is not the case, then further details should be mentioned here for these 2 scale factors.

Yes, we have now clarified this point in the revised manuscript.

Ln. 250 – 260. In your second approach of CO2:CO joint inversion, how do you account for CO2 production from the CO reaction with the radical OH?

In short, we don't. By using fixed OH fields we have effectively linearized the inverse problem. As such, we do not account for this feedback. We have made this point clearer in the revised manuscript.

Ln. 295. Any explanations why the bias is larger with CO than with the CO2-only inversion? It is nowhere mentioned which emissions are optimized in your inversions. Are the emissions from biomass burning optimized as well as biogenic and anthropogenic emissions?

This represents only a small difference very likely due to the introduction of the CO:CO2 correlation.

Biomass burning is included in the optimized combustion term. This is described in section 2.2 on line 159 of the original manuscript.

Ln. 296. Based on which Figure or results do you observed seasonal biases?

These were described but not shown. We have now made that point explicit in the revised manuscript.

Figure 3. Satellite joint inversion shows lower combustion emissions in winter and fall than other months and compared to the other inversions. When looking at Figure A.3, we do observe lower posterior emissions than prior emissions for the same months particularly with satellite observations. Any assumptions why? Are these lower posterior emissions linked to combustion or non-combustion?

During winter CO is dominated by combustion emissions but during summer months it is difficult to determine whether these changes are due to combustion or non-combustion sources.

Figure 3 and 4. The spatial distribution of observations between satellite and in-situ measurements is not the same over Europe with in-situ observations mainly in northern Europe, however your European a posteriori combustion emission seems to match between both set of observations. How do you explain it? And how would this spatial difference impact your inversions?

Most of the emissions over mainland Europe are in northern European countries (Figure 5) where there is the highest density of in situ measurements and where satellite data are available. There are some differences in the resulting combustion CO2 emissions, particularly for the CO:CO2 inversions, and the uncertainties are reasonably large so different estimates are typically not statistically different.

Ln. 324. This sentence would need further information and details. I do not see for the joint in situ inversion an increase for all months and all years in Figure 4. Which increase are you talking about? Like all inversions, we can observe a decrease from January to July and then an increase for the rest of the months. Annual and seasonal variabilities seem to agree between all inversions. We do observe an over-estimation for the join inversion at the annual scale compared to the other inversions but not an increase.

Thanks to the reviewer for spotting this loose language. This text refers to the joint inversion (orange upward triangles) that is lower for the satellite data than for the in situ data. We have made this point clearer in the revised manuscript.

*Ln.* 363. Are we talking about decrease in combustion emissions or an underestimation/reduction in the estimation of the inversions compared to the prior

Point taken. We have reworded this statement in the revised manuscript.

**Reviewer 3**

This study presents a novel CO2 inversion framework that leverages CO and CO2 measurements to estimate CO2 combustion emissions. The framework employs an ensemble Kalman filter technique that incorporates the covariance between CO and CO2 emissions. The integration of CO observations improved the agreement with CO2 observations, and more accurate results are achieved by accounting for grid-scale CO2 error correlations. While the topic is highly relevant and the methodology is sound, the study falls short of demonstrating substantial and statistically significant improvements or changes in emissions at the country scale.

Yes, that's one of our key points. We explore the robustness of our approach but trying to improve on ffCO2 emission estimates over Europe, where there is already good knowledge, is challenging.

The paper would benefit from additional explanation and discussion of both methodology and results. Major revisions are required to address these issues.

**Specific comments:**

NO2 has been extensively used to estimate CO2 combustion emissions. It would be beneficial for the authors to discuss in detail the potential advantages of using CO instead of NO2, including any specific scenarios or conditions where CO may offer better insights or accuracy.

Good idea. We have now included this in the revised manuscript.

Validation of posterior CO concentrations is important to determine the success of CO inversion. The authors should include this validation to justify the impact of adding CO and to provide more credibility to the inversion framework.

We did compare our prior and posterior fields with TROPOMI CO but the improvement was very marginal. Using in situ data, the Pearson correlation coefficient between GEOS-Chem and in situ data increased by 4% for the joint CO2:CO inversion and by 5% for the TNO joint inversion. Using satellite, the correlation coefficient increased by 6% for the joint inversion by 4% for the TNO joint inversion. We have added this information to the revised manuscript.

The paper lacks spatial maps of prior and posteriori CO2 concentrations and comparisons against satellite and in situ observations. Such maps are necessary to visualize the differences and improvements obtained. Furthermore, while the regional comparisons show slight improvements due to the TNO methodology, a more detailed validation, including a time series analysis at each in-situ observation site, would provide more insight into the performance of the methodology.

We did not see significant changes in posterior CO2 concentrations so these maps would not show much beyond what is reported in Figure 7 and Table A2.

The regional comparisons are similar to the results on European scale. There is not sufficient validation to differentiate between the modestly different posterior results.

The term Relative Standard Deviation (RSD) is not defined in the manuscript. The authors need to clarify whether this RSD is based on the analysis spread in the EnKF approach for emissions or concentrations. Due to the different prior uncertainties and inflation applied, careful consideration should be given to the interpretation of the RSD. The authors should discuss whether the RSD provides a meaningful measure of uncertainty reduction in their context.

Thanks for spotting this oversight: we have now defined the metric in the text. The error reduction is a metric that has some value. It is useful to compare the performance of a priori and a posteriori emission to reproduce atmospheric data.

The study reports very small changes in CO2 combustion emissions in general, which raises questions about the accuracy of prior emissions and the effectiveness of the inversion process for Europe. The authors should discuss whether this small change represents a very accurate prior emissions or a potential shortcoming of the inversion methodology.

No, as discussed in the conclusions, our result isn't a shortcoming of the inversion approach per se. Instead, it is a result of applying the approach in an environment that is not currently overwhelmed with satellite data and where we already have excellent knowledge of emission estimates on a country scale.

Given the small changes in CO2 combustion emissions at country scale, it would be insightful to compare their impacts at the sectoral level, especially with the TNO approach. This comparison could highlight any sector-specific discrepancies or improvements.

Also, given the small change in CO2 combustion emissions at the national level, it would be more informative to compare the impact of the TNO approach at the sectoral level. This comparison may highlight improvements needed in bottom-up inventory.

We assume here that the reviewer is asking whether we see an under- or over-estimate of CO2 combustion emissions on the domain wide level. The focus of this study was to inform national inventories and on total combustion emissions rather than attributing any posterior refinements to individual sectors for the EU+UK domain, which is why we didn't discuss it.

In Table 1, we show that the inversions summed across the domain show very little change to the domain scale emissions. But the spatial corrections in Figure 6 show that we consistently see decreases in the urban area around the Germany-Netherlands border. We have a large negative bias in our vehicle emissions in urban areas in the bottom-up inventory, so we think this overestimation is more likely due to residential combustion being overestimated in the bottom-up inventory. However, based on our results we cannot with any confidence attribute these changes to any one sector. The most we can say is that in urban areas – regions with the largest fluxes and the high population densities – we see the largest changes to the prior emissions.

The overall impact on non-combustion emissions is also very small, except for France. The authors need to explore whether the results a high accuracy of prior biospheric fluxes and whether it is consistent with previous inversion studies that adjusted biospheric fluxes only. In addition, an explanation is needed for the large changes in non-combustion emissions in France.

While we have some ability to separate combustion and non-combustion estimates of CO2, the posterior uncertainties only show a modest decrease. The slightly larger improvement of the French estimates reflect the size and distribution of the biospheric fluxes (Figure 2) that are geographically distinct from the largest combustion sources.

Line 84: The statement "few studies have focused on using these data to constrain CO2 flux estimates over mainland Europe or the UK" requires appropriate references to support the claim.

This has been addressed in the revised manuscript.

Line 122: The assumption that the atmosphere is well-mixed when the standard deviation of CO2 concentrations across the lowest five vertical model levels is smaller than 0.3 ppm may not be robust. More meaningful results could be obtained by accounting for potential errors in chemical transport model mixing and by including, for example, PBL height from meteorological data as an additional parameter. Potential errors associated with this assumption should be discussed.

The motivation for this approach is to remove data points for which model error may dominate its interpretation. There is no hard and fast rule for this. On reflection, we think that extending our state vector is unwise given the volume of data being used to constrain the inverse problem.

Line 168: "We use the CAMS fields at their provided temporal resolution (3-hourly) and re-grid to the GEOS-Chem horizontal spatial resolution of 2° x 2.5°": The study uses the regional nested domain of GEOS-Chem with the CAMS fields for lateral boundary conditions. The authors should clarify whether a global domain in GEOS-Chem is required and also describe the performance of the CAMS data as the results presented in this study may be significantly affected by these boundary conditions.

The boundary conditions are updated in the inversion so any biases in these values should be corrected to fit the data.

Line 204: "For CO2, we use an a priori model error of 1.5 ppm for the satellite inversion (Feng et al., 2017) and 3 ppm for the in situ inversion (within the range of Monteil et al., 2020 and White et al., 2019). For CO, we use an a priori model error of 15 and 20 ppb for the satellite and in situ inversions, respectively (Northern Hemisphere CO column and surface mole fraction model- observation differences from Bukosa et al., 2023)": I believe the model errors in the ensemble Kalman filter are estimated from ensemble model simulations. The authors should clarify whether these errors were estimated from something else.

No, at least in our ensemble Kalman filter, an estimate of the model random error is not determined by the ensemble statistics. For our calculations we only perturb the emissions. Perturbing the meteorology in some way would require a GCM not a CTM. Quantifying model error remains a significant challenge for the wider community. This has now been clarified in the manuscript.

Line 207: "For the observations, we use the errors as provided for the satellite or in situ network, averaged to the model resolution.": When using the errors provided for satellite or in situ networks, it is important to consider any error correlations. The manuscript

should address whether such correlations were considered and their potential impact on the results.

We do use off-diagonal elements based on the spatial and temporal proximity of the observations. This is described near the end of section 2.3.

Line 201: "We use an assimilation window of two weeks and a lag window of one month, accounting for the impact of historical emissions on our assimilation period.": The choice of a two-week assimilation window and a one-month lag window could significantly affect the inversion results. A more thorough discussion of the sensitivity of the results to the window size and lag is needed.

We used two weeks to account for temporal variations in emissions. Our previous tests showed that lag periods longer than a month didn't have a major impact because signals decay by that time and, if anything, may introduce spurious correlations.

Line 229: "For our inversions using in situ observations, we localize by distance so that each state vector element that represents a grid-scale variable is only influenced by observations within a 1000 km range.": The manuscript mentions localization by distance for in situ observations but does not clarify if a different setting was applied for satellite observations. This needs to be addressed to understand the methodology.

Our preparatory calculations showed that observations in the immediate vicinity of state vector elements have the most influence. This is explained by typical emission signals decaying exponentially with distance from the point of release. Using observations further away than 1000 km may lead to spurious correlations that impact results, so we used a hard cut-off. We have now clarified this point in the revised manuscript.

Line 236: Applying the scale factor implies the assumption that the location of CO2 sources is perfectly represented by the prior inventory. The authors should provide a more detailed discussion of this assumption and its implications for the inversion results.

Knowledge of the CO2 emissions, particularly over Europe, is not a significant source of error. If we had focused on another region of the world where we have much less knowledge this point would be more valid.

The dimension of the transport scale factor (Eq. 9) is not clear. It should be clearly stated whether it represents only one boundary value or a 3-dimensional distribution.

We solve for four lateral boundary conditions. This is now clearer in the revised manuscript.

Line 260: "For our first two approaches, we assume an a priori uncertainty of 20%...": The balance of adjustments for transport, chemistry, and combustion uncertainties in the obtained results has not been discussed. This could have a significant impact on inversion results and requires careful consideration.

Estimating the uncertainties of these individual terms builds on previous work by us and others in the fields. We have used the best knowledge we have at the time of writing. The biggest challenge we face is the small uncertainties associated with the combustion emission that effectively prevent a substantial revision in CO2 emissions over Europe. This is discussed in section 4.

Line 276: "We increase the provided uncertainties by a factor of 3 to make them more comparable with our other simulations.": There must be a clear justification for this increase. The authors need to explain the rationale for this adjustment and its impact.

The scale factor was chosen based on consultation with the inventory developers based on the "max error scenario" they were willing to allow given current knowledge.

Line 278: "This results in a mean grid-scale CO2 combustion uncertainty of 18%, though there is greater variability in grid cell uncertainties compared to our other approaches.": Inclusion of spatial patterns of these uncertainties would be useful in visualizing their distribution and impact on CO2 emissions estimates.

With respect, we disagree. All we are saying is that the mean uncertainty for this approach is larger than the others tested in this study. But the variation in uncertainties between grid boxes can be larger for the other methods. Given the modest improvements we get using the largest mean uncertainty values, we don't think adding this figure would add much to the manuscript.